# DEX-AR: A Dynamic Explainability Method for Autoregressive Vision-Language Models

## Abstract

As Vision-Language Models (VLMs) become increasingly sophisticated and widely used, it becomes more and more crucial to understand their decision-making process. Traditional explainability methods, designed for classification tasks, struggle with modern autoregressive VLMs due to their complex token-by-token generation process and intricate interactions between visual and textual modalities. We present DEX-AR (Dynamic Explainability for AutoRegressive models), a novel explainability method designed to address these challenges by generating both per-token and sequence-level 2D heatmaps highlighting image regions crucial for the model's textual responses. The proposed method offers to interpret autoregressive VLMs—including varying importance of layers and generated tokens—by computing layer-wise gradients with respect to attention maps during the token-by-token generation process. DEX-AR introduces two key innovations: a dynamic head filtering mechanism that identifies attention heads focused on visual information, and a sequence-level filtering approach that aggregates per-token explanations while distinguishing between visually-grounded and purely linguistic tokens. Our evaluation on ImageNet, VQAv2, and PascalVOC, shows a consistent improvement in both perturbation-based metrics, using a novel normalized perplexity measure, as well as segmentation-based metrics. [1]

## 1 Introduction

Vision-Language Models (VLMs) have emerged as a transformative force in AI, with remarkable capabilities in bridging visual understanding and natural language generation. Recent advances have produced increasingly better models like LLaVA (Liu et al., 2024b;a), PaliGemma (Beyer et al., 2024), Gemini-1.5 (Team, 2024) and GPT-4o (Achiam et al., 2023), which can engage in complex visual reasoning tasks ranging from image captioning to open-ended dialogue about visual content. These models have found applications across diverse domains, from assisting visually impaired users (Roslyn et al., 2024; Yang et al., 2024) to enabling more natural human-AI interaction through multimodal interfaces (Zhao et al., 2024b). However, as these models grow in complexity and capability, understanding their decision-making process becomes increasingly challenging. Modern VLMs employ architectures that combine visual encoders with LLMs through multiple layers of attention mechanisms, making it difficult to trace how visual information influences generated text. Understanding this process is crucial, as interpretability studies (Adly Templeton, 2024; Guan et al., 2023) have revealed critical failure modes in VLMs and shown that model explanations significantly improve human-AI collaboration. This understanding becomes particularly important as VLMs are deployed in high-stakes applications like autonomous systems (Buçinca et al., 2021).

While numerous explainability methods exist for computer vision models (Selvaraju et al., 2020; Chefer et al., 2021a;b) and language models( (Zini & Awad, 2022; Zhao et al., 2024a)), these approaches fall short when applied to modern autoregressive VLMs. Traditional computer vision explainability methods typically focus on classification tasks with fixed outputs, failing to capture the dynamic nature of token-by-token generation and the complex interaction between visual and textual modalities in VLMs. Furthermore, while significant progress has been made in explaining contrastive models (e.g., CLIP) (Gandelsman et al., 2024; 2025; Abnar & Zuidema, 2020; Chefer et al., 2021a;b; Bousselham et al., 2025), these approaches are designed to account for the dynamic nature

---

[1]All code and scripts will be publicly released upon acceptance.

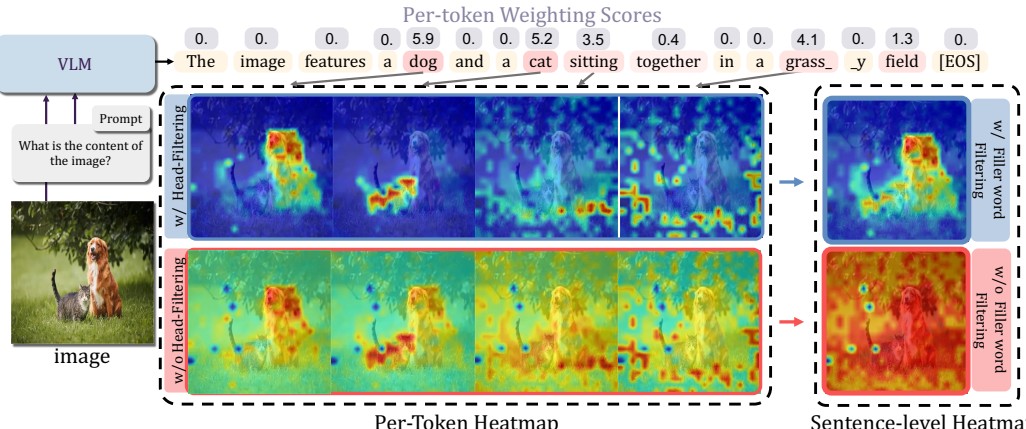

Figure 1: Example of token-level and sentence-level attribution maps for Vision-Language Models (VLMs). Given an input image and prompt, DEX-AR produces per-token heatmaps highlighting relevant image regions for each generated word. These are then aggregated into a final sentence-level heatmap using token-specific weighting scores that reflect visual relevance.

of autoregressive generation. Although emerging methods like TAM (Li et al., 2025) have begun to address this, accurately capturing the varying importance of different generated tokens remains challenging, as some serve primarily linguistic functions while others directly reference visual content. This gap suggests that directly applying traditional methods to autoregressive tasks may lead to incomplete interpretations, highlighting the need for specialized approaches that address both visual grounding and sequential dependencies This lack of suitable explainability methods could led to potentially misleading interpretations from traditional approaches, inhibiting the improvement of model reliability as well as the detection potential failure modes.

We argue that the unique characteristics of autoregressive VLMs demand a specialized approach to explainability. Namely, as these models generate text tokens sequentially, with each token potentially attending to different parts of the image and previous textual context, understanding this process requires tracking the flow of visual information through the model's layers and identifying which image regions influence specific generated tokens—a challenge not addressed by current explainability methods. To fill this gap, we introduce DEX-AR (Dynamic Explainability for AutoRegressive models), a novel explainability method specifically designed for autoregressive VLMs. DEX-AR leverages layer-wise gradients with respect to attention maps to produce 2D heatmaps that highlight the image regions most influential for each generated token. On top of that, we apply a dynamic head filtering mechanism that identifies attention heads focused on visual information, and a token-level filtering approach that distinguishes between visually-grounded and purely linguistic tokens. The resulting framework provides fine-grained per-token explanation maps that uncover the model's decision-making process at each generation step, enabling a layer-wise analysis to understand information flow throughout the network, while remaining model-agnostic by leveraging the attention's gradient, common to all transformer-based architectures.

We evaluate the proposed method on various downstream tasks and datasets, including perturbation on ImageNet (Russakovsky et al., 2015) and VQAv2 (Goyal et al., 2017), showing a consistent improvement over alternative explainability methods across different VLM architectures. To further quantitatively evaluate the proposed filtering, we introduce PascalVOC-QA, a specialized dataset that provides natural language question-answer pairs with segmentation information and explicit annotations distinguishing between tokens derived from visual content and linguistic filler tokens. It shows that the dual-filtering approach effectively distinguishes visually-relevant content, improving the Signal-to-Noise Ratio from 9.16 to 96.12 on Pascal-QA.

We summarize the contributions of this work as follows:

1) We propose a gradient-based explainability method specifically designed for autoregressive VLMs, handling the specific characteristics of token-by-token generation.

2) We extend the proposed method by a general dual-filtering mechanism that dynamically weighs attention heads and tokens based on their visual relevance.

3) We propose a new evaluation setup together with a set of metrics to assess the quality of the explainability methods for autoregressive VLMs.

## 2 RELATED WORKS

**Explainability in Deep Learning Models** Recent years have witnessed significant advances in explaining deep learning models' decisions, particularly in computer vision tasks. Traditional gradient-based methods such as Grad-CAM (Selvaraju et al., 2020), Guided Backpropagation (Springenberg et al., 2015), and Integrated Gradients (Sundararajan et al., 2017) compute gradients of target outputs with respect to input features or intermediate activations to generate saliency maps. While effective for convolutional neural networks (CNNs) in classification tasks, these methods face limitations when applied to modern transformer-based architectures (Vaswani, 2017) and sequential generation tasks (Karpathy et al., 2016). Attention mechanisms have emerged as an alternative approach to model interpretability (Clark et al., 2019; Galassi et al., 2019). Methods like Attention Roll-out (Abnar & Zuidema, 2020) aggregate attention weights across layers to trace information flow. However, recent studies have shown that attention weights alone may not reliably indicate feature importance (Wiegreffe & Pinter, 2019; Serrano & Smith, 2019), particularly in deep architectures where attention patterns are complex (Brunner et al., 2020). This limitation is especially pronounced in multimodal settings where visual and textual information interact (Li et al., 2022; 2023; Liu et al., 2024b;a; Xiao et al., 2024; Beyer et al., 2024). **Explainability in Vision-Language Models** VLMs present unique challenges for explainability due to their multimodal nature and complex architectures. A substantial body of work has focused on interpreting contrastive models like CLIP. Methods such as (Gandelsman et al., 2024; Bousselham et al., 2025) and adaptations of generic transformer explainability (Chefer et al., 2021a) have proven effective in visualizing how these models align image and text embeddings. However, these techniques are primarily designed for static alignment tasks and do not account for the dynamic state changes inherent to autoregressive generation. Existing methods have been adapted for VLMs, such as developing hybrid approaches that combine gradients with attention (Chefer et al., 2021b;a; Barkan et al., 2023). Furthermore, established techniques such as Integrated Gradients (IG) (Sundararajan et al., 2017) and model-agnostic approaches like RISE (Petsiuk et al., 2018) could also been utilized to interpret VLMs. However, several key challenges remain unaddressed in current approaches:**Sequential Generation:** Most existing methods are designed for fixed outputs and struggle with the token-by-token generation process in autoregressive models (Liu et al., 2024b; Beyer et al., 2024; Liu et al., 2024a; SkunkworksAI, 2024; Team, 2024; Achiam et al., 2023). **Token-Level vs Sequence-Leven Attribution:** Current approaches either lack the granularity to attribute model outputs at the level of individual tokens, or fail to distinguish between content and filler words. Recent work has attempted to address these challenges through various approaches. Methods like (Ding et al., 2017) have explored visualizing information flow in neural machine translation, while others have focused on developing token-level attribution techniques (Ferrando et al., 2022). Most recently, TAM (Li et al., 2025) proposed an estimated causal inference method to mitigate context interference in autoregressive VLMs. However, TAM relies on static visual features and post-hoc statistical estimation. In contrast, DEX-AR utilizes layer-wise gradients to capture the dynamic attention mechanism at each specific generation step. DEX-AR, builds upon these foundations while extending those ideas by a novel layer-wise gradient computation approach combined with dynamic filtering mechanisms, enabling attribution of visual information throughout the autoregressive generation process.

## 3 METHOD

### 3.1 AUTOREGRESSIVE VISION-LANGUAGE MODELS

Current VLMs such as LLaVA combine a visual encoder — typically a Vision Transformer (ViT) — with a Large Language Model (LLM) to process multimodal content and generate text.

Let $N$ denote the number of visual tokens produced by the visual encoder, $T_c$ the number of context tokens in the input prompt, and $T_a$ the number of answer tokens generated autoregressively by the

model. The visual encoder processes an input image to generate a set of $N$ visual tokens, which are concatenated with $T_c$ context tokens that encapsulate instructions or queries, thereby forming an initial token sequence. This sequence, comprising $N + T_c$ tokens, is ingested by the LLM.

The LLM consists of $L$ transformer layers, each equipped with $h$ attention heads, and generates a textual response comprising $T_a$ tokens. At each generation step $t \in \{1, \ldots, T_a\}$, the model processes a token sequence of length $T_t = N + T_c + t$, where $t$ denotes the current step in the autoregressive process. Consequently, the total number of tokens processed by the LLM upon completion of generation is $T = N + T_c + T_a$.

Formally, for each layer $l \in \{1, \ldots, L\}$ and generation step $t$, the LLM produces hidden states denoted as $Z^{l,t} \in \mathbb{R}^{T_t \times d}$, where $d$ is the embedding dimension. For clarity and without loss of generality, we assume that the first $N$ tokens correspond to the visual tokens and the subsequent $T_c$ tokens correspond to the context tokens, such that:

$$Z^{l,t} = \{ \underbrace{z_1^l, \ldots, z_N^l}_{\text{visual tokens } Z_v^l}, \underbrace{z_{N+1}^l, \ldots, z_{N+T_c}^l}_{\text{context tokens } Z_c^l}, \underbrace{y_1^l, \ldots, y_t^l}_{\text{answer tokens } Y_a^l} \}$$
$$= \{Z_v^l, Z_c^l, Y_a^{l,t}\}. \tag{1}$$

Here, $Z_v^l \in \mathbb{R}^{N \times d}$ and $Z_c^l \in \mathbb{R}^{T_c \times d}$ denote the visual and context tokens, respectively, which remain invariant across generation steps due to the causal attention mechanism employed by the LLM.

At each step $t$, within each layer $l$, the model computes attention maps $A^{l,t} \in \mathbb{R}^{h \times T_t \times T_t}$, where $T_t = N + T_c + t$, capturing the interaction weights across the $h$ attention heads. At every generation step $t$, the LLM outputs a logits vector $o^t \in \mathbb{R}^V$, where $V$ is the vocabulary size, via a linear projection of the final hidden state:

$$o^t = \text{LM\_Head}(Z_{-1}^{L,t}) = \text{LM\_Head}(y_t^L) \tag{2}$$

Then based on these logits, typically by applying a softmax function followed by a selection mechanism such as argmax or sampling the predicted word is selected. We denote $\hat{o}^t \in R$ the logit corresponding to the selected word in the vocabulary. This autoregressive process generates the answer, with each token generation step influenced by both visual, context and already generated answer, as encoded in the hidden states and attention maps.

## 3.2 EXPLAINABILITY PER TOKEN

Based on the objective of computing explainability maps that highlight the influence of regions of the image on each generated token in the autoregressive process, we leverage gradients w.r.t. the attention maps in the transformer layers of the LLM. The method operates as follows:

**Computing Intermediate Logits:** At each generation step $t \in \{1, \ldots, T_a\}$, we aim to assess the influence of the visual tokens $Z_v^l$ on the prediction of the next token. To isolate the contributions from each layer $l \in \{1, \ldots, L\}$, we compute the intermediate logits using the hidden states from layer $l$ instead of the final output layer. Specifically, we obtain:

$$o^{l,t} = \text{LM\_Head}(Z_{-1}^{l,t}) = \text{LM\_Head}(y_t^l) \in \mathbb{R}^V \tag{3}$$

where $Z_{-1}^{l,t}$ is the hidden state corresponding to the last token (the one being predicted) at layer $l$. We denote $\hat{o}^{l,t} \in \mathbb{R}$ as the logit corresponding to the sampled word at step $t$. This operation follows the *Logit Lens* approach (nostalgebraist, 2020), which interprets intermediate hidden states by projecting them into the vocabulary space to reveal the model's prediction confidence at varying depths. This specific conditioning on the last token index is structurally dictated by the causal attention mechanism inherent to autoregressive Transformers. Due to the causal masking, the hidden state at the current step $t$ serves as the sole information bottleneck accumulating the context of all preceding tokens $(1 \ldots t)$ and visual embeddings required for the next-token prediction $P(y_{t+1}|y_{1:t}, I)$. Consequently, computing logits and gradients from this state isolates the specific decision boundary for the current generation step, whereas conditioning on previous indices would redundantly probe fixed historical predictions.

**Gradient Computation:** We compute the gradient of the logit $\hat{o}^{l,t}$ w.r.t. the attention map $A^{l,t} \in \mathbb{R}^{h \times T_t \times T_t}$ at layer $l$: $\nabla A^{l,t} = \frac{\partial \hat{o}^{l,t}}{\partial A^{l,t}} \in \mathbb{R}^{h \times T_t \times T_t}$ where $h$ is the number of attention heads and $T_t = N + T_c + t$ is the total number of tokens processed up to step $t$.

**Focusing on the Last Token and Visual Tokens:** Since we are interested in the generation of the current token, we focus on the gradient w.r.t. the attention maps involving the last token in the sequence. We extract the gradients corresponding to the last row (see Figure 2): $\nabla A^{l,t}_{-1} = \nabla A^{l,t}[:, -1, :] \in \mathbb{R}^{h \times T_t}$. Next, we isolate the gradients of the visual tokens: $\nabla A^{l,t}_{-1,v} = \nabla A^{l,t}_{-1}[:, :N] \in \mathbb{R}^{h \times N}$ where $N$ is the number of visual tokens.

**Dynamic Head Filtering:** In practice, not all heads/layers contribute equally to attending to the visual tokens during the generation process. Some attention heads may focus primarily on textual information or other aspects that are not directly relevant to the image content.

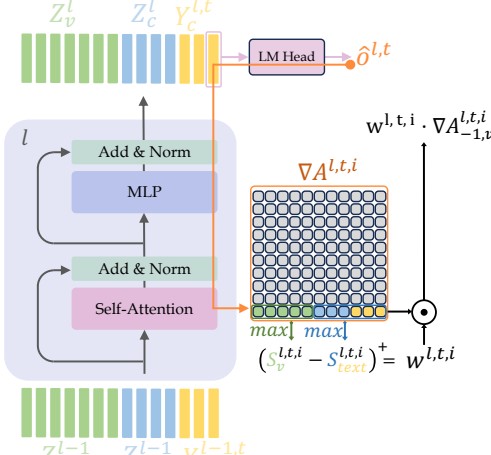

Figure 2: Architecture overview of DEX-AR. At each layer $l$, head $i$ and generation step $t$, gradients of attention maps are computed and weighted based on their relative focus on visual versus textual tokens to produce attribution maps.

Including gradients from such heads can introduce noise into the explainability maps, reducing their interpretability. To address this, we introduce a filtering mechanism that dynamically weighs the contributions of different attention heads and layers based on their relative focus on visual tokens. For each attention head $i \in \{1, \ldots, h\}$ at layer $l$ and generation step $t$, we compute the maximum gradient magnitude with respect to the visual tokens and the text tokens (i.e., context and answer tokens):

$$S^{l,t,i}_{\text{img}} = \max_j \left( \nabla A^{l,t}_{-1,v}[i,j] \right) \in R, \quad S^{l,t,i}_{\text{text}} = \max_j \left( \nabla A^{l,t}_{-1,\text{text}}[i,j] \right) \in R, \tag{4}$$

where $\nabla A^{l,t}_{-1,v}[i,:]$ represents the gradients with respect to visual tokens and $\nabla A^{l,t}_{-1,\text{text}}[i,:]$ represents the gradients with respect to text tokens for head $i$ (see Figure 2).

The weighting factor for each attention head is computed as: $w^{l,t,i} = \left( S^{l,t,i}_{\text{img}} - S^{l,t,i}_{\text{text}} \right)^+$, where $(\cdot)^+$ denotes the ReLU function, ensuring that only positive differences contribute. This weighting factor reflects the focus of the attention head on visual tokens relative to text tokens. A higher $w^{l,t,i}$ indicates that head $i$ at layer $l$ is more attentive to the image content than to the text. A maximum-based approach is particularly effective here because it captures the strongest visual attention signals regardless of object size. For instance, when localizing a small object like a "tennis ball" versus a large object like "sky", averaging-based approaches would bias towards larger objects due to their spatial extent. In contrast, a maximum-based approach identifies the most salient visual signals independent of their coverage, leading to more accurate localization across objects of varying sizes (see also Sec.4.5). The heatmap for the generated token $t$, noted $E^{(t)}$, is then a weighted sum of:

$$\bar{E}^{(t)} = \overbrace{\sum_{l=1}^{L} \underbrace{\sum_{i=1}^{h} w^{l,t,i} \cdot \nabla A^{l,t}_{-1,v}}_{over\ heads}}^{over\ layers} \in R^N, \quad E^{(t)} = \text{Norm}\left( \text{Reshape}(\bar{E}^{(t)}) \right) \in R^{W \times H}, \tag{5}$$

where Norm is the min-max normalization, Reshape reshape the tokens to a 2D grid and $H$ and $W$ are the height and width dimensions of the visual token grid.

## 3.3 SEQUENCE-LEVEL EXPLAINABILITY MAP

When VLMs generate descriptions or answer questions about images, they produce sequences of tokens that vary in their reliance on visual information. Understanding the model's reasoning requires

aggregating these token-level explanations while accounting for their varying visual relevance. For example, in the response "The young woman is wearing a floral dress", tokens like "woman", "floral", and "dress" directly reference visual content, while "the", "is", and "wearing" serve primarily grammatical functions. We ground this filtering in the interpretation of gradients as a measure of prediction sensitivity. A high gradient magnitude implies that perturbing the attending features would significantly degrade model confidence for the current token. By comparing the maximum sensitivity to visual features ($S_{img}^l$) against textual history ($S_{text}^l$), $\delta^t$ quantifies the "visual necessity" of the generated token. This effectively filters out tokens governed by linguistic priors, where the model is robust to visual perturbation, while amplifying those where visual evidence is a necessary condition for prediction. To address this, we introduce an additional filtering mechanism that weighs the contribution of each generated token based on its reliance on visual information, see Fig. 1. For each token $t$, we compute a token-specific weight by comparing the maximum attention to visual versus textual content across all heads and layers: $\delta^t = \left( \max_{l,i} S_{img}^{l,t,i} - \max_{l,i} S_{text}^{l,t,i} \right)^+$, where $\max_{l,i}$ denotes the maximum over all layers $l$ and heads $i$. This weighting scheme effectively suppresses the contribution of tokens that are primarily predicted based on linguistic context rather than visual information. The final explainability map for the whole sequence of $T_a$ generated tokens is:

$$
\bar{E} = \overbrace{\sum_{t=1}^{T_a} \delta^t \cdot E^{(t)}}^{\text{over tokens}} \in R^N, \quad E = \texttt{Norm}\left(\texttt{Reshape}(\bar{E})\right) \in R^{W \times H}. \tag{6}
$$

This dual-filtering approach at both head resp. layer level and token level ensures that the explainability maps focus on tokens and attention patterns that are genuinely informed by visual content, rather than those driven by linguistic context.

## 4 EXPERIMENTS

### 4.1 DATASETS AND TASKS

**Perturbation:** ◇**Task:** To quantitatively evaluate the proposed method, we employ a perturbation-based evaluation protocol. This approach is particularly valuable for assessing explainability methods as it directly measures how the removal of supposedly important image regions impacts model performance. By systematically perturbing regions identified as significant, it can be verify whether these regions truly contribute to the model's decision-making process. Specifically, for a range of percentages $p \in \{0\%, 10\%, ..., 90\%\}$, we replace the top-$p$ pixels according to the heatmap values with the dataset's mean pixel value. ◇**Metric:** While such evaluation protocols are well-established for classification tasks, they present unique challenges for generative models like VLMs. Traditional metrics such as classification accuracy are not directly applicable due to the open-ended nature of text generation and the complexity of comparing generated text sequences. Prior work often relies on additional LLMs or manual evaluation to assess the correctness of generated responses, which can introduce external biases and typically yields only binary success metrics. We propose a more natural approach leveraging the model's internal confidence through perplexity measurements. For a sequence of ground truth tokens $y = (y_1, ..., y_T)$, the perplexity is computed as: $\text{PPL}(y) = \exp\left( \frac{1}{T} \sum_{t=1}^{T} -\log P(y_t | y_{<t}, \mathcal{I}) \right)$ where $\mathcal{I}$ represents the (potentially perturbed) input image. Higher perplexity scores indicate increased uncertainty in the model's predictions, making it an effective measure of how perturbations affect the model's confidence and accuracy. When important visual regions are removed, we expect to see a significant increase in perplexity, reflecting the model's reduced ability to generate accurate responses without crucial visual information. This metric provides a continuous score reflecting how confidently the model predicts the expected sequence of tokens. To account for varying baseline perplexities across different samples and model capacities, we normalize the perplexity at each perturbation level by the original (unperturbed) perplexity: $\text{PPL}^{\text{norm}}(p) = \frac{\exp(\text{PPL}(y|\mathcal{I}_p))}{\exp(\text{PPL}(y|\mathcal{I}_0))}$ where $\mathcal{I}_p$ denotes the image with $p\%$ of pixels perturbed. This normalization helps isolate the impact of the perturbation from the inherent difficulty of predicting certain sequences. The final evaluation metric is computed as the Area Under the Curve (AUC) of the normalized perplexity versus perturbation percentage: $\text{AUC} = \int_0^{0.9} \text{PPL}^{\text{norm}}(p) dp$ For positive perturbation, a higher AUC indicates that perturbing pixels identified as important by the

heatmap lead to a larger degradation in model performance, suggesting a more accurate attribution map. We evaluate all methods using both positive perturbation (removing highest-valued pixels) and negative perturbation (removing lowest-valued pixels) to ensure a comprehensive assessment of the attribution quality. For a detailed discussion of perplexity as evaluation metric, we refer to the appendix, Sec F. ◇**Datasets:** We compare the performance on two dataset: ImageNet (Russakovsky et al., 2015) and VQAv2 (Goyal et al., 2017). From the ImageNet-val, we randomly sampled 5,000 images spanning 1,000 object categories.From VQAv2-val, we randomly sampled 1,000 image-question pairs spanning various question types (e.g., yes/no, counting, open-ended).This diverse selection ensures a comprehensive evaluation of all method across recognition tasks.

**Segmentation-based Evaluation** ◇**Task:** While perturbation analysis provides insights into the overall quality of the resulting attributions, we also evaluate how precisely explicitly referenced objects can be localized. This evaluation serves as a critical sanity check: when a VLM identifies specific objects in an image, the attribution map should highlight the corresponding regions. ◇**Metrics:** To quantitatively assess localization accuracy, we employ two complementary metrics. First, we compute the Intersection over Union (IoU) between the continuous attribution maps and ground truth masks. Given the continuous nature of attribution maps $A$, we determine the optimal threshold $\tau^*$ for each prediction by maximizing the IoU score across $k = 20$ thresholds: $\text{IoU}(A, M) = \max_{\tau \in \mathcal{T}} \frac{|\{A > \tau\} \cap M|}{|\{A > \tau\} \cup M|}$ where $M$ represents the ground truth binary mask and $\mathcal{T}$ is a set of $k$ equally spaced thresholds spanning the range of attribution values. We also introduce a soft version of IoU that directly operates on continuous-valued attribution maps without requiring thresholding: $\text{Soft-IoU}(A, M) = \frac{\sum_{i,j} A_{i,j} M_{i,j}}{\sum_{i,j} A_{i,j} + \sum_{i,j} M_{i,j} - \sum_{i,j} A_{i,j} M_{i,j}}$.

This formulation provides a smooth evaluation metric that does not depend on thresholding, thus avoids the potential loss of information from binary thresh. Additionally, we employ the Energy Pointing Game (EPG) metric, which evaluates how well the total attribution energy aligns with the target object without requiring threshold selection: $\text{EPG}(A, M) = \frac{\sum_{i,j} A_{i,j} M_{i,j}}{\sum_{i,j} A_{i,j}} \times 100$. This metric measures the percentage of the total attribution energy within the ground truth mask, providing a threshold-free assessment of localization accuracy. ◇**Dataset:** We evaluate these metrics on the Pascal VOC dataset, which provides object segmentation masks across 20 common object categories. For each image, we prompt the VLM with a simple classification query (e.g., `"Classify the image"`) and evaluate the attribution map against the ground truth segmentation masks of all objects present in the image. This setup ensures that the model must identify and localize multiple objects simultaneously, providing a more challenging and realistic evaluation scenario. We report IoU and EPG metrics as complementary metrics: while IoU assesses the spatial alignment of the attributions with ground truth segments, EPG measures how well the method concentrates attribution energy on relevant objects.High performance on both metrics indicates that a method can reliably identify regions important to the model's visual reasoning process.

**Filler-words Filtering Evaluation** To quantitatively evaluate the dual-filtering strategy, we construct PascalVOCQA, a specialized dataset derived from PascalVOC that enables quantitative assessment of filler word detection. For each image, a controlled natural language description is generated where objects are connected using predefined filler phrases (e.g., *"I see a {object 1} as well as a {object 2}"*). By constructing the expected answer in a systematic way, with predefined start phrases (e.g., *"I see"*) and connecting phrases (e.g., *"as well as"*), we maintain token-level annotations of which parts of the response correspond to filler words versus content-bearing tokens. This automated construction provides ground truth annotations for the position of the "filler words" in the sequence, enabling quantitative evaluation of the filtering mechanism's ability to distinguish between tokens that convey visual content and those that serve linguistic functions.

## 4.2 EXPERIMENTAL SETUP

**Implementation Details.** We evaluate the method on diverse state-of-the-art Vision-Language Models (VLMs) with varying architectural designs. We first consider decoder-only models: LLaVA-1.5 (Liu et al., 2024a), which combines a CLIP ViT-L/14 (Radford et al., 2021) vision encoder with Vicuna, and BakLLaVA (SkunkworksAI, 2024), which adopts a similar architecture but uses Mistral-7B (Jiang et al., 2023) as its foundation. We also evaluate on PaliGemma (Beyer et al.,

| Model | Method | ImageNet Pos.(↑) | ImageNet Neg.(↓) | VQAv2 Pos.(↑) | VQAv2 Neg.(↓) | (↓)sec./img. | Model | Method | ImageNet Pos.(↑) | ImageNet Neg.(↓) | VQAv2 Pos.(↑) | VQAv2 Neg.(↓) |
|---|---|---|---|---|---|---|---|---|---|---|---|---|
| LlaVA-1.5 | Attention | 2.17 | 1.01 | 0.88 | 0.89 | 0.45 | Florence2 | Attention | 0.78 | 0.84 | 0.94 | 1.00 |
| | GradCAM | 1.43 | 1.1 | 0.91 | **0.77** | 0.66 | | Rollout | 0.82 | 0.80 | 0.94 | 0.98 |
| | CheferCAM | 2.06 | 1.03 | **0.93** | 0.78 | 6.20 | | CheferCAM | 0.81 | 0.82 | **0.97** | 0.97 |
| | Attn×Grad | 1.94 | 1.00 | 0.90 | 0.80 | 0.65 | | Attn×Grad | 0.80 | 0.81 | 0.95 | 0.99 |
| | Int.Grad | 1.63 | 1.52 | 0.91 | 0.80 | 8.20 | | Int.Grad | 0.84 | 0.80 | 0.76 | **0.74** |
| | RISE | 1.24 | 0.99 | 0.92 | 0.78 | 11.8 | | RISE | 0.83 | **0.78** | 0.89 | 0.89 |
| | IIA | 1.66 | 0.90 | 0.92 | 0.77 | 15.3 | | IIA | 0.79 | 0.80 | 0.95 | 0.99 |
| | DEX-AR | **2.31** | **0.96** | **0.93** | 0.77 | 0.71 | | DEX-AR | **0.85** | 0.78 | **0.97** | 0.95 |
| BakLLaVA-v1 | Attention | 11.47 | 4.36 | 0.98 | 0.86 | 0.45 | PaliGemma | Attention | 1.71 | 1.45 | 0.94 | 0.75 |
| | Rollout | 6.13 | 3.12 | 0.95 | 0.90 | 0.51 | | Rollout | 1.68 | 1.49 | 0.90 | 0.80 |
| | GradCAM | 7.49 | 4.39 | 1.01 | 0.86 | 0.66 | | GradCAM | 1.74 | **0.87** | 0.85 | 0.87 |
| | CheferCAM | 11.15 | 4.07 | 1.05 | 0.84 | 6.20 | | CheferCAM | 1.71 | 1.44 | 0.94 | **0.74** |
| | Attn×Grad | 12.6 | 3.74 | 1.06 | 0.86 | 0.65 | | Attn×Grad | 1.74 | 1.32 | **0.96** | 0.75 |
| | Int.Grad | 13.50 | 12.77 | 1.03 | 0.84 | 8.20 | | Int.Grad | 1.86 | 1.48 | 0.86 | 0.86 |
| | RISE | 6.39 | 3.87 | 1.09 | 0.86 | 11.8 | | RISE | 1.63 | 1.04 | 0.95 | 0.79 |
| | IIA | 11.08 | 3.77 | 1.02 | 0.86 | 15.3 | | IIA | 1.75 | 1.40 | 0.93 | 0.76 |
| | DEX-AR | **18.10** | **2.48** | **1.13** | **0.81** | 0.71 | | DEX-AR | **2.71** | 0.90 | 0.95 | 0.75 |

Table 1: **Perturbation-based evaluation** across different VLM architectures (Decoder-only, Encoder-Decoder, Prefix-Decoder) on ImageNet and VQAv2. Values represent Area Under the perplexity Curve (AUC) for positive (↑) and negative (↓) perturbations.

2024), which employs a prefix language modeling approach allowing bidirectional attention between visual and textual tokens, and Florence-2 (Xiao et al., 2024), which utilizes cross-attention mechanisms for multimodal integration. This diverse selection allows us to validate DEX-AR's effectiveness across different VLM architectures.

**Baselines:** We compare DEX-AR against a comprehensive set of explainability methods, including gradient-based (GradCAM (Selvaraju et al., 2020), Attn×Grad (Chefer et al., 2021b), Integrated Gradient (Sundararajan et al., 2017), CheferCAM (Chefer et al., 2021a) and IIA (Barkan et al., 2023)), attention-based (Raw Attention, Attention Rollout (Abnar & Zuidema, 2020)) and perturbation-based (RISE (Petsiuk et al., 2018)), as detailed in Table 1. This diverse set spans established paradigms, enabling thorough validation of DEX-AR's effectiveness. Implementation details are provided in the Annex.

## 4.3 PERTURBATION RESULTS

Table 1 compares DEX-AR against established explainability methods across different VLM architectures. Note that metric ranges vary across models due to base performance differences, making relative improvements more meaningful. ◇On ImageNet, DEX-AR shows superior performance across architectures, notably on BakLLaVA-v1 (AUC of 18.1 for positive perturbation, 5.5 points above Attn×Grad) while achieving better negative perturbation scores (2.48), indicating accurate identification of both relevant and irrelevant regions. On the more complex VQAv2 dataset, DEX-AR maintains higher positive perturbation scores (1.13 for BakLLaVA-v1, 0.95 for PaliGemma), though with more modest gains. On PaliGemma, while achieving the best positive scores (2.71), it performs similarly to baselines in negative perturbation (0.87 vs GradCAM's 0.90). Moreover, DEX-AR is significantly faster to traditional methods like Integrated Gradient, RISE, IIA or CheferCAM. DEX-AR's consistent performance across architectures (Decoder-only, Encoder-Decoder, and Prefix-Decoder) validates its approach for understanding VLMs' visual reasoning.

Table 2: **Segmentation Performance on PascalVOC.** Comparison of different methods on four VLM architectures using soft-IoU and IoU, and EPG (Energy Pointing Game).

| Model | Method | soft-IoU(↑) | IoU(↑) | EPG(↑) |
|---|---|---|---|---|
| LlaVA-1.5 | Attention | 2.00 | 19.10 | 16.00 |
| | Rollout | 2.54 | 19.59 | 11.41 |
| | GradCAM | 10.20 | 28.90 | 19.30 |
| | CheferCAM | 1.60 | 21.01 | 17.10 |
| | Attn×Grad | 5.10 | 24.20 | 26.60 |
| | DEX-AR | **17.70** | **36.34** | **27.75** |
| BakLLaVA-v1 | Attention | 2.04 | 19.56 | 16.84 |
| | Rollout | 3.30 | 20.14 | 14.67 |
| | GradCAM | 8.19 | 24.36 | 24.76 |
| | CheferCAM | 1.62 | 21.07 | 17.07 |
| | Attn×Grad | 5.09 | 24.23 | 26.32 |
| | DEX-AR | **17.00** | **35.85** | **26.33** |
| Florence2 | Attention | 8.37 | 21.20 | 20.51 |
| | Rollout | 5.67 | 26.59 | 21.34 |
| | CheferCAM | 9.68 | 21.16 | 18.87 |
| | Attn×Grad | 9.35 | 23.92 | 23.15 |
| | DEX-AR | **17.10** | **32.48** | **24.46** |
| PaliGemma | Attention | 4.11 | 20.38 | 16.55 |
| | GradCAM | 8.44 | 22.55 | **26.95** |
| | CheferCAM | 4.61 | 21.05 | 18.02 |
| | Attn×Grad | 6.55 | 23.32 | 23.52 |
| | DEX-AR | **15.26** | **23.55** | 20.43 |

## 4.4 Segmentation Results

Table 2 compares the localization performance of DEX-AR to various explainability baselines across multiple state-of-the-art VLMs. When applied to LLaVA-1.5, DEX-AR achieves substantial improvements over existing methods, with a soft-IoU of 17.70%, IoU of 36.34%, and EPG of 27.75% – representing relative improvements of 73.5%, 25.7%, and 4.3% respectively over the next best method. This performance advantage is consistently maintained across different model architectures, including BakLLaVA-v1 and Florence2, where DEX-AR outperforms traditional approaches like Attention, Rollout, and GradCAM by significant margins. The method's effectiveness is particularly evident in the soft-IoU metric, where it achieves scores approximately 2-3 times higher than conventional approaches, indicating superior ability to produce continuous attribution maps that align with ground truth object segments. Notably, while the performance slightly decreases for PaliGemma, DEX-AR still maintains its lead over baseline methods, suggesting that the layer-wise gradient approach effectively captures the model's reasoning process across different VLM architectures.

## 4.5 Dynamic Filtering Ablations

**Heads Filtering:** We conduct an ablation study to evaluate the effectiveness of the attention head filtering mechanism, which aims to identify and prioritize attention heads that are more focused on visual information. We compare different filtering approaches: using only the maximum gradient value (max), selecting a percentage of top gradient values ($k_\%$), or averaging all gradients (avg). We evaluate these variants using two complementary metrics: Signal-to-Noise Ratio (SNR) and Mean Squared Error (MSE). SNR measures the ratio between the explanation intensity inside versus outside the ground truth mask: $\text{SNR} = 10 \cdot \log\left(\frac{\langle E,M\rangle/\|M\|_1}{\langle E,(1-M)\rangle/\|1-M\|_1}\right)$ where $E \in \mathbb{R}^{H\times W}$ is the explanation map, $M \in \{0,1\}^{H\times W}$ is the binary ground truth mask, $\langle\cdot,\cdot\rangle$ denotes the element-wise dot product, and $\|\cdot\|_1$ is the L1 norm. Higher SNR values indicate better localization as they reflect stronger activation inside the target region compared to the background. MSE quantifies the direct alignment between the explanation map and the ground truth mask, where lower values indicate better agreement.

Table 3: **Head Filtering Ablation:** Analysis of different filtering thresholds ($k_\%$) measuring Signal-to-Noise Ratio (SNR) and Mean Squared Error (MSE) on PascalVOC.

| | LlaVA | | BakLlaVA | |
|---|---|---|---|---|
| $k_\%$ | SNR($\uparrow$) | MSE($\downarrow$) | SNR($\uparrow$) | MSE($\downarrow$) |
| - | 1.64 | 0.33 | 5.27 | 0.15 |
| max | **3.64** | **0.22** | **6.02** | **0.14** |
| 0.05 | 3.35 | 0.24 | 4.13 | 0.17 |
| 0.1 | 3.09 | 0.25 | 4.13 | 0.19 |
| 0.2 | 2.45 | 0.28 | 3.18 | 0.22 |
| 0.3 | 1.98 | 0.31 | 2.36 | 0.25 |
| 0.4 | 1.66 | 0.33 | 1.92 | 0.26 |
| 0.5 | 1.45 | 0.34 | 1.62 | 0.27 |
| 0.6 | 1.29 | 0.35 | 1.42 | 0.28 |
| 0.7 | 1.20 | 0.35 | 1.30 | 0.29 |
| 0.8 | 1.16 | 0.36 | 1.20 | 0.29 |
| 0.9 | 1.12 | 0.36 | 1.14 | 0.29 |
| avg | 1.09 | 0.30 | 1.05 | 0.30 |

Results in Table 3 demonstrate that selective filtering of attention heads significantly improves explanation quality across both models. The max-based filtering achieves the best performance, improving SNR from 1.64 to 3.64 for LLaVA and from 5.27 to 6.02 for BakLLaVA, while also reducing MSE. Using more gradient values to compute the head scoring ($k_\%$) consistently degrades performance, suggesting that considering too many gradients introduces noise in the head selection process. This is further evidenced when using the average of all gradients (avg) to score the heads, which performs even worse than the baseline (no filtering), highlighting the importance of selective head filtering based on the most salient gradient signals.

Table 4: **Filler words Filtering Ablation:** Analysis of the impact of the different filtering strategies on the filtering of the filler words produced by the VLM on PascalVOC using LlaVA-1.5-7B.

| Filtering | | SNR($\uparrow$) | MSE($\downarrow$) | EPG($\uparrow$) |
|---|---|---|---|---|
| Head | Filler | | | |
| ✗ | ✗ | 9.16 | 0.13 | 44.96 |
| ✓ | ✗ | 16.84 | **0.10** | 63.38 |
| ✗ | ✓ | 89.29 | 0.11 | 92.50 |
| ✓ | ✓ | **96.12** | 0.12 | **95.04** |

**Filler words Filtering:** Using PascalVOC-QA with LLaVA-1.5-7B, we assess the impact of both, head-level filtering, which weighs attention heads based on their visual focus, and filler word filtering, which identifies tokens primarily driven by linguistics.

We employ three complementary metrics: Signal-to-Noise Ratio (SNR) measuring the ratio between attention given to non-filler words versus the attention given to filler words; Mean Squared Error (MSE) quantifying the pixel-wise deviation from ground truth masks; and Energy Point Game (EPG) evaluating the proportion of explanation energy that falls within the ground truth regions.

Results in Table 4 demonstrate the benefits of the dual-filtering approach. While head filtering alone improves performance across all metrics compared to the baseline (SNR: $9.16 \rightarrow 16.84$, EPG: $44.96\% \rightarrow 63.38\%$), the most substantial gains come from filler word filtering.

### 4.6 QUALITATIVE EXAMPLES

Figure 3 compares DX-AR against to baselines across diverse queries, showing focused attributions that align with the objects being discussed, while comparable methods tend to generate more diffuse or scattered heatmaps. The "no Filtering" row particularly highlights the advantage of the dual-filtering approach, producing clean, object-centric attributions where other methods struggle to distinguish between relevant objects and backgrounds. Further qualitative results can be found in Section J.

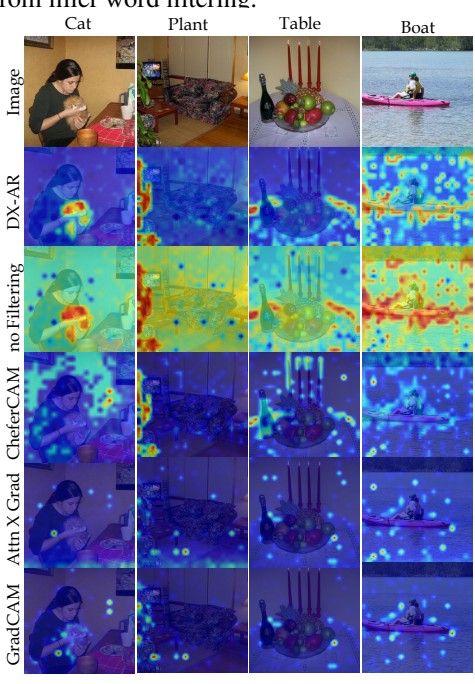

Figure 3: **Qualitative Comparison**

### 5 CONCLUSION

We presented DEX-AR, a novel explainability method for autoregressive VLMs that introduces dynamic head and token-level filtering mechanisms, demonstrating substantial improvements over existing methods across multiple architectures. The dual-filtering approach validates the importance of selective attention for accurate attribution, while providing a tool for better model understanding and facilitating the responsible deployment of AI systems.

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

APPENDIX

## A OVERVIEW

In this supplementary material, we first provide implementation details in section B, including model versions and prompt templates used across our experiments. Section D details the baseline methodologies, presenting formal definitions and adaptations of existing explainability approaches for autoregressive Vision-Language Models. We describe our extension of DEX-AR to models with cross-attention mechanisms in section E, explaining the necessary architectural adaptations. Section F provides a comprehensive rationale for selecting perplexity as our primary evaluation metric, contrasting it with traditional NLP metrics. Additional evaluation protocols are documented in section G, including insertion/deletion metrics and the Perplexity Information Curve. Section H presents ablation studies that validate our key design choices, particularly the use of ReLU activation and intermediate layer representations. Finally, sections I and J offer insights into our dataset construction process and qualitative examples that illustrate both the strengths and limitations of our approach across different model architectures.

## B IMPLEMENTATION DETAILS

For our experiments, we utilized several state-of-the-art Vision-Language Models (VLMs) accessed through the HuggingFace Transformers library. To ensure reproducibility, we specify the exact model weights and prompt templates used for each architecture.

**Model Versions** We employed the following model checkpoints:

- LLaVA: `llava-hf/llava-1.5-7b-hf`
- BakLLaVA: `llava-hf/bakLlava-v1-hf`
- PaLiGemma: `google/paligemma-3b-pt-224`
- Florence-2: `microsoft/Florence-2-base`

**Prompt Templates** Each model was trained with specific prompt formatting conventions, which we strictly adhered to following the guidelines in their respective papers:

- For LLaVA and BakLLaVA: "`USER: <image>\nClassify the image. ASSISTANT:`"
- For PaLiGemma: "`Cap.`"
- For Florence-2, we utilized the `<DETAILED_CAPTION>` template, which is automatically converted to "`Describe in detail what is shown in the image.`" by the tokenizer.

## C COMPARISON WITH TOKEN-SPECIFIC ATTENTION MAPPING (TAM)

In this section, we provide a comparative analysis between DEX-AR and the recently proposed Token-specific Attention Mapping (TAM). While both methods aim to elucidate the token generation process in VLMs, they rely on fundamentally different mechanisms to attribute importance.

The core distinction lies in the treatment of the autoregressive state. TAM uses a "Logit Lens", computing relevance maps by projecting cached visual states $\mathbf{F}^v$ onto the vocabulary space via the token embedding $\mathbf{w}_{token}$ (i.e., $\mathbf{F}^v \cdot \mathbf{w}_{token}$). Crucially, in causal transformer, these visual states $\mathbf{F}^v$ remain invariant throughout the generation process, which introduces some limitations. For example, in scenarios involving repeated class labels, since $\mathbf{F}^v$ is static, the initial heatmap calculation for the token "cat" is mathematically identical regardless of its position in the sequence, see Figure4. To address this, TAM relies on heuristic post-processing, including the subtraction of accumulated maps and Rank Gaussian Filtering (RGF) to smooth high-frequency "salt-and-pepper" noise. While effective

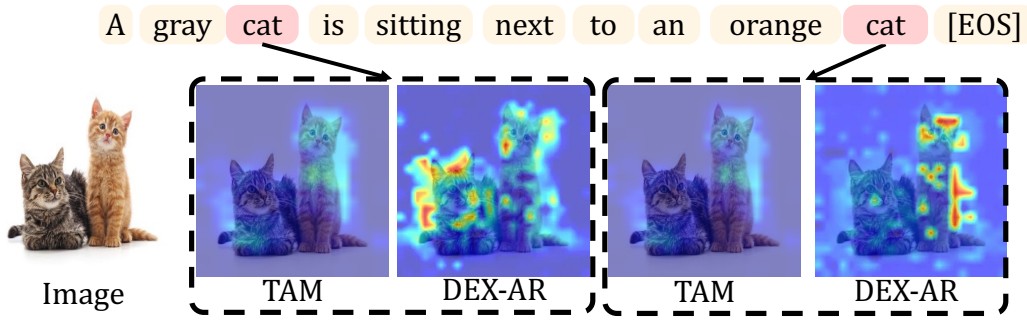

Figure 4: **Qualitative comparison between TAM and DEX-AR.** Given the prompt "A gray cat is sitting next to an orange cat [EOS]", we visualize the attribution maps for the first occurrence of "cat" (referring to the gray cat) and the second (referring to the orange cat). DEX-AR produces sharp, instance-specific localizations that accurately reflect the distinct visual grounding of each token. In contrast, TAM produces more diffuse maps, relying on static feature projections and heuristic filtering.

for visualization, these smoothing operations impose prior structures that may not necessarily reflect the model's intrinsic decision-making process.

In contrast, DEX-AR computes gradients with respect to attention maps at the specific generation step $t$. This approach captures the active query-key interactions within the self-attention mechanism at that precise moment. As illustrated in Figure 4, when the model generates the sequence "A gray cat... an orange cat", DEX-AR naturally distinguishes between the two instances—highlighting the gray cat at the first occurrence and the orange cat at the second. By leveraging the dynamic nature of the gradient flow rather than static feature projections, DEX-AR faithfully disentangles distinct object instances without relying on heuristic smoothing or post-hoc subtraction.

## D    BASELINES IMPLEMENTATION DETAILS

### D.1    GRADCAM BASELINE

GradCAM (Selvaraju et al., 2020), originally designed for CNN architectures, can be adapted to VLMs by applying gradient-based attribution to transformer hidden states. For each generated token $t \in \{1, \ldots, T_a\}$, the method computes gradients with respect to the hidden states at a specific layer $l$. Unlike our approach which leverages attention patterns, GradCAM focuses directly on the hidden state representations:

$$E^t = \text{ReLU}\left(\frac{\partial \hat{o}^{L,t}}{\partial Z_v^l} \cdot Z_v^l\right) \cdot \mathbf{1}^T \in \mathbb{R}^N \tag{7}$$

where $Z_v^l \in \mathbb{R}^{N \times d}$ represents the hidden states of visual tokens at layer $l$, $\hat{o}^{L,t}$ is the logit of the predicted token at the final layer $L$, and $\mathbf{1}$ is a vector of ones that sums over the embedding dimension. The final explainability map aggregates the gradients over all generated tokens:

$$E = \text{Norm}\left(\text{Reshape}\left(\sum_{t=1}^{T_a} E^t\right)\right) \in \mathbb{R}^{W \times H} \tag{8}$$

This adaptation of GradCAM to VLMs maintains the core principle of using gradients to identify important features, but operates on transformer hidden states rather than CNN feature maps. Note that GradCAM computes gradients with respect to the final model output, which differs from our layer-specific approach.

## D.2 CheferCAM Baseline

CheferCAM (Chefer et al., 2021b) extends the principles of GradCAM to transformer architectures by propagating relevance through attention layers. Unlike our approach which computes layer-specific gradients, CheferCAM computes gradients with respect to the final model output only. For each generated token $t \in \{1, \ldots, T_a\}$, the method computes a relevance matrix $R^t \in \mathbb{R}^{T_t \times T_t}$, initialized as an identity matrix. The relevance is then propagated through the layers by combining attention maps with their gradients computed with respect to the final logit $\hat{o}^{L,t}$:

$$R^{t,0} = I_d \in \mathbb{R}^{T_t \times T_t}$$

$$R^{t,l} = R^{t,l-1} + \text{ReLU}\left(\sum_{i=1}^{h} A_i^{l,t} \odot \frac{\partial \hat{o}^{L,t}}{\partial A_i^{l,t}}\right) \cdot R^{t,l-1} \tag{9}$$

$$R^t = R^{t,L}$$

where $A_i^{l,t} \in \mathbb{R}^{T_t \times T_t}$ is the attention map for head $i$ at layer $l$, $I_d$ is the identity matrix, $\hat{o}^{L,t}$ is the logit of the predicted token at the final layer $L$, and $\odot$ denotes the element-wise multiplication. The final explainability map is obtained by extracting the relevance scores corresponding to the visual tokens for the last generated token and aggregating over all generated tokens:

$$E = \text{Norm}\left(\text{Reshape}\left(\sum_{t=1}^{T_a} R_{-1,:N}^{t,L}\right)\right) \in \mathbb{R}^{W \times H} \tag{10}$$

where $R_{-1,:N}^{t,L}$ selects the relevance scores between the last token and the visual tokens.

## D.3 Attn×Grad

Attn×Grad, originally proposed as a baseline in (Chefer et al., 2021b), provides an alternative approach that directly leverages both the raw attention maps and their gradients. We extend their single-layer formulation to incorporate multiple layers, similar to our DEX-AR approach, therefore providing a fairer comparison by allowing the method to leverage information across the entire network depth. For each generated token $t \in \{1, \ldots, T_a\}$ and layer $l$, the method computes:

$$E^{t,l} = \text{ReLU}\left(\sum_{i=1}^{h} A_{i,-1,v}^{l,t} \odot \frac{\partial \hat{o}^{l,t}}{\partial A_{i,-1,v}^{l,t}}\right) \in \mathbb{R}^N \tag{11}$$

where $A_{i,-1,v}^{l,t} \in \mathbb{R}^N$ represents the attention weights from the last token to visual tokens for head $i$, $\hat{o}^{l,t}$ is the logit of the predicted token computed from layer $l$'s hidden states, and $\odot$ denotes element-wise multiplication. The final explainability map aggregates the gradients over all layers and generated tokens:

$$E = \text{Norm}\left(\text{Reshape}\left(\sum_{t=1}^{T_a}\sum_{l=1}^{L} E^{t,l}\right)\right) \in \mathbb{R}^{W \times H} \tag{12}$$

While this approach shares our intuition about leveraging attention gradients across multiple layers, it lacks the dynamic head filtering and token-level weighting mechanisms that allow DEX-AR to better distinguish visually-relevant information.

## D.4 Attention Rollout Baseline

Attention Rollout (Abnar & Zuidema, 2020) provides a gradient-free approach to compute attention-based explanations by recursively combining attention maps across layers. For each generated token $t \in \{1, \ldots, T_a\}$, the method first averages attention weights across heads for each layer:

$$\bar{A}^{l,t} = \frac{1}{h}\sum_{i=1}^{h}A_i^{l,t} + I \in \mathbb{R}^{T_t \times T_t} \tag{13}$$

where $I$ is the identity matrix added to account for residual connections. The attention weights are then normalized:

$$\hat{A}^{l,t} = \text{Normalize}(\bar{A}^{l,t}) = \frac{\bar{A}^{l,t}}{\sum_j \bar{A}_{ij}^{l,t}} \tag{14}$$

The final attention matrix is computed by recursively multiplying the normalized attention matrices from all layers:

$$R^t = \prod_{l=L}^{1}\hat{A}^{l,t} \in \mathbb{R}^{T_t \times T_t} \tag{15}$$

The explainability map is then obtained by extracting the attention weights from the last generated token to the visual tokens and aggregating over all generated tokens:

$$E = \text{Norm}\left(\text{Reshape}\left(\sum_{t=1}^{T_a}R_{-1,:N}^t\right)\right) \in \mathbb{R}^{W \times H} \tag{16}$$

While this method provides a computationally efficient approach by avoiding gradient computation, it lacks the ability to capture the direct influence of attention patterns on the model's predictions.

### D.5 RAW ATTENTION BASELINE

As a simple baseline, we consider directly using the raw attention weights without any gradient information or recursive propagation. For each generated token $t \in \{1, \ldots, T_a\}$ and layer $l$, we average the attention weights across heads:

$$E^{t,l} = \frac{1}{h}\sum_{i=1}^{h}A_{i,-1,v}^{l,t} \in \mathbb{R}^N \tag{17}$$

where $A_{i,-1,v}^{l,t}$ represents the attention weights from the last token to visual tokens for head $i$. The final explainability map aggregates these attention weights across all layers and generated tokens:

$$E = \text{Norm}\left(\text{Reshape}\left(\sum_{t=1}^{T_a}\sum_{l=1}^{L}E^{t,l}\right)\right) \in \mathbb{R}^{W \times H} \tag{18}$$

This naive approach serves as a control to demonstrate that more complex processing of attention patterns, such as our gradient-based methods, is necessary for meaningful explanations. While raw attention weights might capture some notion of token relationships, they fail to account for how these attention patterns actually influence the model's predictions.

## E  DEX-AR FOR CROSS-ATTENTION

**Background on Cross-Attention:** Some Vision-Language Models (VLMs) employ cross-attention mechanisms to fix information between visual and textual modalities. In encoder-decoder architectures like Florence-2, the process occurs in two stages: first, the visual tokens ($Z_v$) and context tokens ($Z_c$) are processed in the encoder, producing fused representations. These representations then serve as keys and values for the decoder's cross-attention layers, while the generated tokens ($Y_a$) act as queries.

| ReLU | Interm. feat. | LLaVA | BakLLaVA | PaliGemma | Florence-2 |
|:---:|:---:|:---:|:---:|:---:|:---:|
| ✗ | ✓ | 31.56 | 35.09 | 22.90 | 30.44 |
| ✓ | ✗ | 33.05 | 32.87 | 22.85 | 26.91 |
| ✗ | ✗ | 24.54 | 27.80 | 22.83 | 27.01 |
| ✓ | ✓ | **36.34** | **35.85** | **23.55** | **32.48** |

Table 5: Ablation study results showing IoU scores on PascalVOC across different model architectures. "ReLU" indicates the use of ReLU activation on attention gradients, and "Interm. feat." indicates the use of intermediate layer features to compute the gradient.

Formally, for a given layer $l$ and generation step $t$, the cross-attention operation can be expressed as:

$$\text{CrossAttn}(Q, K, V) = \text{softmax}\left(\frac{QK^T}{\sqrt{d}}\right)V \tag{19}$$

where $Q \in \mathbb{R}^{t \times d}$ represents queries from the decoder (generated tokens), and $K, V \in \mathbb{R}^{(N+T_c) \times d}$ are keys and values derived from the encoder's output (combined visual and context tokens).

**Gradient Computation with Cross-Attention:** For encoder-decoder models like Florence-2, we compute gradients with respect to the cross-attention maps $A^{l,t} \in \mathbb{R}^{h \times t \times (N+T_c)}$ at each layer $l$ and generation step $t$. The process follows these steps:

1. Compute intermediate logits for the current token:

$$\hat{o}^{l,t} = \text{LM\_Head}(y_t^l) \in \mathbb{R} \tag{20}$$

2. Calculate gradients with respect to cross-attention maps:

$$\nabla A^{l,t} = \frac{\partial \hat{o}^{l,t}}{\partial A^{l,t}} \in \mathbb{R}^{h \times t \times (N+T_c)} \tag{21}$$

3. Extract gradients for the last token's attention to visual tokens:

$$\nabla A_{-1,v}^{l,t} = \nabla A^{l,t}[:, -1, : N] \in \mathbb{R}^{h \times N} \tag{22}$$

The autoregressive nature of the generation process is handled similarly to self-attention. The key differences in implementation are:

- **Encoder-Decoder Models:** We compute gradients with respect to cross-attention maps between the decoder and the combined visual-textual representations from the encoder.
- **Decoder-Only Models:** We compute gradients with respect to self-attention maps, focusing on attention patterns between generated tokens and visual tokens in the same attention layer.

The core methodology remains consistent across architectures, with the primary adaptation being the type of attention maps analyzed (cross-attention vs. self-attention) and the location of visual tokens in the attention computation.

# F RATIONALE FOR PERPLEXITY AS AN EVALUATION METRIC

In this section, we elaborate on the rationale for selecting perplexity as the primary metric to evaluate the performance of explainability methods for autoregressive VLMs. We contrast this choice with traditional natural language processing (NLP) metrics such as CIDEr, SPICE, and BERT-Score, demonstrating why perplexity is better suited for our perturbation-based evaluation protocol and why alternative metrics fall short in this context.

Our evaluation is centered around assessing how perturbations to image regions identified by the explainability method's heatmaps affect the model's ability to predict a fixed ground truth answer.

Unlike generative tasks where new text is produced and compared to references, our setup focuses on the model's internal confidence in reproducing a predefined sequence of tokens. Perplexity, defined as the exponential of the average negative log-likelihood of the ground truth tokens, directly quantifies this confidence:

$$\text{PPL}(y) = \exp\left(\frac{1}{T}\sum_{t=1}^{T} -\log P(y_t|y_{<t}, \mathcal{I})\right), \tag{23}$$

where $y = (y_1, \ldots, y_T)$ is the ground truth sequence and $\mathcal{I}$ is the input image. When critical image regions are masked, a significant increase in perplexity indicates that the model struggles to predict the correct sequence, validating the heatmap's accuracy in identifying visually influential areas. This direct alignment with our evaluation goal–measuring the impact of visual perturbations—makes perplexity a natural fit.

To enhance robustness, we employ a normalized metric: the ratio of perplexity on the perturbed image to perplexity on the unperturbed image:

$$\text{PPL}^{\text{norm}}(p) = \frac{\text{PPL}(y|\mathcal{I}_p)}{\text{PPL}(y|\mathcal{I}_0)}, \tag{24}$$

where $\mathcal{I}_p$ denotes the image with $p\%$ of pixels perturbed. This relative change isolates the effect of masking specific regions, mitigating variations in baseline perplexity due to linguistic complexity or inherent sample difficulty. For instance, some ground truth answers may naturally exhibit higher perplexity due to rare vocabulary or syntactic structure. By normalizing, we ensure that the metric reflects the perturbation's impact rather than absolute prediction difficulty, enabling fair comparisons across diverse examples in datasets like ImageNet and VQAv2.

In contrast, traditional NLP metrics such as CIDEr, SPICE, and BERT-Score are ill-suited for this task. These metrics are designed to evaluate the quality of generated text against reference texts, typically in open-ended generation scenarios. Applying them here would require generating new text from perturbed images and comparing it to the ground truth, which diverges from our objective. We are not assessing the quality or semantic similarity of a generated output but rather the model's confidence in a fixed output under varying visual inputs. For example, a VLM might describe an image of a "coucal" as "a bird" after perturbation–a factually correct but imprecise response. CIDEr, reliant on n-gram overlap, would penalize this heavily. Also, based on initial experiments, the BERT-Score, based on contextual embeddings, overly rewards generic descriptions, even from blurred images, failing to capture the loss of specific visual grounding.

Further challenges arise with these metrics due to the autoregressive nature of VLMs and the diversity of our evaluation datasets. ImageNet, with its 1,000 fine-grained classes, exemplifies this issue: exact class names are rarely produced verbatim by VLMs, especially under perturbation. Metrics requiring precise string matching or reference-based scoring become impractical or misleading. Additionally, BERT-Score's sensitivity to semantic similarity can inflate scores for vague outputs, undermining its ability to reflect the model's reliance on detailed visual features. Perplexity, conversely, operates independently of reference text generation, focusing solely on the model's token-by-token prediction confidence, which aligns seamlessly with the autoregressive process and our perturbation-based approach.

In summary, perplexity offers the following benefits:

- **Direct Measurement of Confidence:** Perplexity quantifies the model's uncertainty in predicting fixed ground truth tokens, directly reflecting the impact of perturbing key image regions identified by DEX-AR.
- **Normalization for Robustness:** The ratio of perturbed to unperturbed perplexity accounts for baseline variations, isolating the perturbation's effect across diverse samples.
- **Incompatibility of NLP Metrics:** Metrics like CIDEr, SPICE, and BERT-Score assess generated text quality, not model confidence in a fixed output, making them misaligned with our evaluation goal.
- **Limitations of Reference-Based Metrics:** VLMs often produce imprecise or generic descriptions (e.g., "a bird" for "coucal"), rendering exact-match metrics unreliable and semantic metrics overly permissive.

| Method | POS↑ | NEG↓ | IC↑ | Insert.↓ | Del.↑ | sec./img.↓ |
|--------|------|------|-----|----------|-------|------------|
| Rollout | 6.13 | 3.12 | 0.69 | 0.14 | 0.21 | 0.51 |
| GradCAM | 7.49 | 4.39 | 0.64 | 0.20 | 0.28 | 0.66 |
| CheferCAM | 11.15 | 4.07 | 0.64 | 0.19 | 0.36 | 6.20 |
| Attn x CAM | 12.60 | 3.74 | 0.65 | 0.18 | 0.42 | 0.65 |
| Integrated Grad. | 13.50 | 12.77 | 0.44 | 0.32 | 0.33 | 3.25 |
| RISE | 6.39 | 3.87 | 0.65 | 0.21 | 0.28 | 11.80 |
| IIA | 11.08 | 3.77 | 0.65 | 0.18 | 0.40 | 15.30 |
| DEX-AR | **18.10** | **2.48** | **0.72** | **0.13** | **0.50** | 0.71 |

Table 6: Comprehensive evaluation of attribution methods on ImageNet using BakLlaVA-7B. POS and NEG refer to positive and negative perturbation metrics from the main paper, IC represents the PIC metric, Insert. and Del. are the insertion and deletion metrics, and sec./img. shows computational efficiency.

- **Continuous Sensitivity:** As a continuous metric, perplexity captures subtle confidence changes, offering a nuanced evaluation of perturbation effects.

- **Practical Efficiency:** Perplexity computation is straightforward and scalable, requiring no additional models or manual intervention, unlike reference-based alternatives.

## G   ADDITIONAL EVALUATION METRICS

To complement our primary perturbation-based evaluation protocol described in the main paper, we introduce three additional metrics that provide a more comprehensive assessment of attribution quality. These metrics evaluate different aspects of how well the attribution method identifies truly important image regions.

### G.1   INSERTION AND DELETION METRICS

**Insertion Metric:**   The insertion metric measures how well the attribution method identifies sufficient regions for model prediction. Unlike the perturbation test that progressively removes important pixels, this metric starts with a heavily blurred image (Gaussian blur with $\sigma = 50$) that preserves only global structure. For increasing percentages $p \in \{0\%, 10\%, ..., 90\%\}$, we reveal the top-$p\%$ pixels identified by the attribution map, replacing blurred pixels with their original values.

We measure the model's perplexity at each step, expecting it to progressively decrease as more relevant information is revealed. Similar to our primary metric, we normalize the perplexities and compute the Area Under the Curve (AUC). A lower AUC score indicates that revealing the pixels identified as important leads to faster recovery of the model's performance, suggesting a more accurate attribution map.

**Deletion Metric:**   The deletion metric provides a complementary evaluation by measuring how quickly the model's performance degrades when progressively removing the most important pixels. Starting with the original image, we iteratively replace the top-$p\%$ most important pixels (according to the attribution scores) with the dataset's mean pixel value for increasing percentages $p \in \{0\%, 10\%, ..., 90\%\}$.

As with the insertion metric, we measure the model's perplexity at each degradation step and compute the normalized AUC. For deletion, a higher AUC indicates that removing the identified important regions causes a more rapid deterioration in the model's performance, suggesting that the attribution method effectively identified the critical regions for the model's prediction.

Together, insertion and deletion metrics provide a comprehensive evaluation: deletion verifies that removing important regions significantly impairs the model's ability to generate accurate responses, while insertion confirms that these regions alone are sufficient to enable the model's prediction.

### G.2 Perplexity Information Curve (PIC)

To provide a more nuanced evaluation of attribution quality, we introduce the Perplexity Information Curve (PIC), adapted from the Precision Information Curves family of metrics. While our primary perturbation metric measures the impact of removing salient regions, PIC evaluates how efficiently the attribution method identifies the minimal image regions necessary for maintaining model performance.

The metric progressively reveals image regions based on their attribution scores, starting from a fully obscured image (all pixels set to dataset mean). At each revelation step, we measure both the information content of the partially revealed image using WebP compression entropy and the model's perplexity on the ground truth answer. These measurements are normalized relative to the fully obscured and original image values:

$$
H_{\text{norm}}(p) = \frac{H(I_p) - H(I_{\text{blur}})}{H(I_{\text{orig}}) - H(I_{\text{blur}})}
$$

$$
\text{PPL}_{\text{norm}}(p) = \frac{\text{PPL}(y|I_p) - \text{PPL}(y|I_{\text{blur}})}{\text{PPL}(y|I_{\text{orig}}) - \text{PPL}(y|I_{\text{blur}})}
$$

where $H(I)$ denotes the compression entropy of image $I$, and $p$ represents the fraction of revealed pixels. The PIC plots normalized perplexity against normalized entropy, with the area under this curve (PIC-AUC) serving as the final metric. A higher PIC-AUC indicates that the attribution method more efficiently identifies regions that are crucial for the model's prediction, achieving better performance with less information revealed.

### G.3 Comprehensive Evaluation Results

Table 6 presents a comprehensive comparison of our DEX-AR method against baseline approaches across all evaluation metrics. The results demonstrate that DEX-AR consistently outperforms existing methods across multiple dimensions of attribution quality:

The results show that DEX-AR achieves the best performance across all attribution quality metrics while maintaining competitive computational efficiency. Specifically, DEX-AR demonstrates superior performance in identifying regions that are both necessary (highest POS and Del. scores) and sufficient (lowest Insert. score) for the model's predictions, while also providing the most information-efficient attributions (highest IC score).

## H Ablation Studies on Design Choices

We conduct an ablation study on the PascalVOC dataset to validate our key design choices in DEX-AR, specifically examining the impact of (1) the ReLU activation applied to the attention gradients and (2) the use of intermediate layer representations versus only the last layer. We report the Intersection over Union (IoU) metric for all experiments.

### H.1 Effect of ReLU on Gradient Attribution

We first investigate the impact of applying ReLU activation to the attention gradients $\nabla A_{-1,v}^{l,t}$. The motivation behind using ReLU is to focus on positive contributions in the gradient attribution maps, effectively filtering out negative gradients that might correspond to inhibitory or contradictory signals. We compare two variants of gradient processing:

$$
\nabla \hat{A}_{-1,v}^{l,t} = \begin{cases} (\nabla A_{-1,v}^{l,t})^+ & \text{with ReLU} \\ \nabla A_{-1,v}^{l,t} & \text{without ReLU} \end{cases} \tag{25}
$$

Our experiments across multiple VLM architectures demonstrate that applying ReLU to the gradients consistently improves performance. As shown in Table 5, using ReLU without intermediate features improves the IoU score from 24.54 to 33.05 on LLaVA, an increase of 8.51 IoU points.

This improvement is consistent across different architectures, with PaliGemma showing an increase of 0.65 IoU points and Florence-2 showing a gain of 2 IoU points. The effectiveness of ReLU can be attributed to its ability to isolate positive attribution signals, leading to more focused and interpretable explanation maps.

## H.2 IMPACT OF INTERMEDIATE LAYER REPRESENTATIONS

We also examine the effectiveness of incorporating gradients from intermediate layers versus using only the last layer. When using intermediate layers, we aggregate gradients across all layers as described in Eq. 5, while the last-layer-only variant computes gradients solely from the final transformer layer:

$$\bar{E}^{(t)} = \begin{cases} \sum_{l=1}^{L} \sum_{i=1}^{h} w^{l,t,i} \cdot \nabla A_{-1,v}^{l,t} & \text{all layers} \\ \sum_{i=1}^{h} w^{L,t,i} \cdot \nabla A_{-1,v}^{L,t} & \text{last layer only} \end{cases} \tag{26}$$

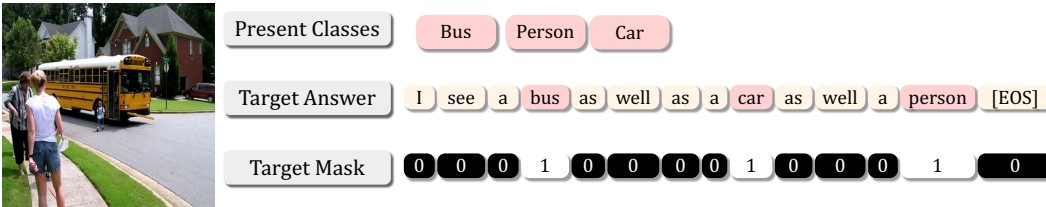

Figure 5: **PascalVOC-QA:** example of the dataset used to evaluate the quality of the filtering.

The results in Table 5 show that using intermediate features consistently improves performance across all architectures. For instance, without ReLU activation, incorporating intermediate features improves the IoU score from 24.54 to 31.56 on LLaVA, an increase of 7.02 IoU points.

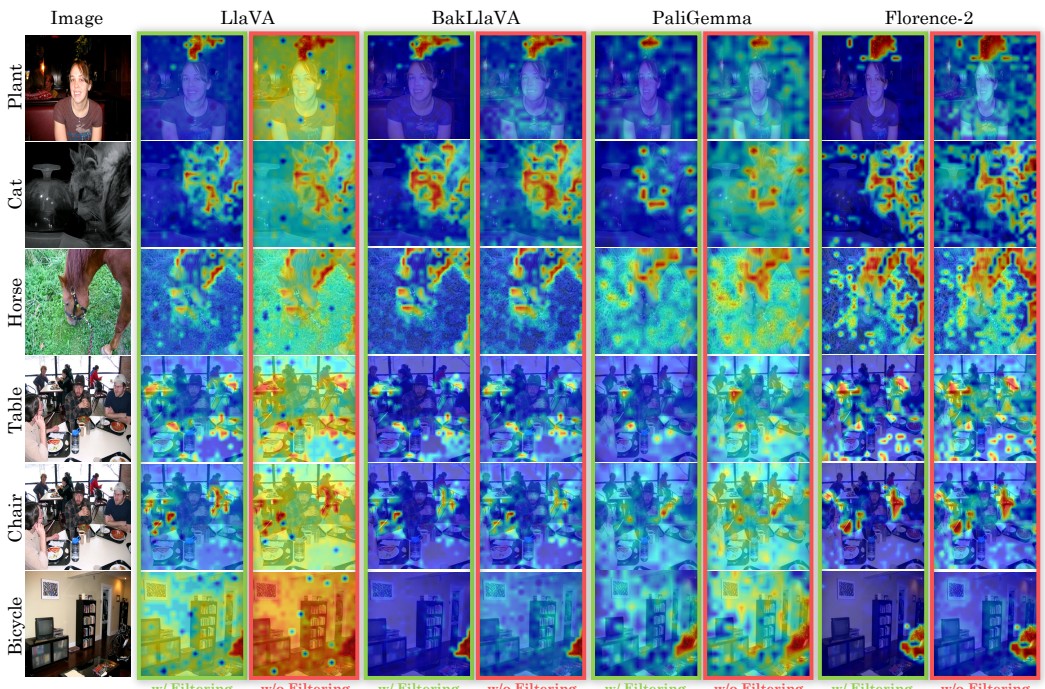

Figure 6: Qualitative examples of the heatmap generated by DEX-AR for different VLMs with and without filtering.

## H.3 Synergistic Effects

Most notably, we observe a synergistic effect when combining both ReLU activation and intermediate layer representations. This combination achieves the best performance across all tested architectures, with LLaVA showing an IoU score of 36.34, an improvement of 11.80 IoU points over the baseline (no ReLU, last layer only). This synergy suggests that the ReLU activation effectively filters relevant attention gradients at each layer, while the intermediate representations capture complementary aspects of the model's visual reasoning process.

The consistent performance improvements across different architectures (LLaVA, BakLLaVA, PaliGemma, and Florence-2) demonstrate the robustness and generalizability of our design choices. These results validate our hypothesis that both selective gradient filtering through ReLU and the incorporation of intermediate layer information are valuable for generating accurate and meaningful attribution maps in autoregressive VLMs.

## I   PascalVOC-QA Dataset Generation.

Figure 5 illustrates the construction process of our PascalVOC-QA dataset. For each image in PascalVOC, we generate a controlled natural language description by combining the ground truth object classes with predefined connecting phrases. The example shows an image containing a bus, a person, and a car, for which we construct the target answer `"I see a bus as well as a car as well as a person [EOS]"`. We also maintain a target mask that indicates which tokens correspond to content-bearing words (1) versus filler words (0). This binary mask enables quantitative evaluation of our filtering mechanism's ability to distinguish between tokens that convey visual content (e.g., "bus", "car", "person") and those that serve purely grammatical functions (e.g., "I", "see", "as", "well"). The dataset's systematic construction ensures consistent evaluation across different images while maintaining natural language structure.

## J   Qualitative Analysis

**Qualitative example**   Figure 6 presents qualitative examples of DEX-AR's explainability maps across different VLMs, demonstrating its effectiveness and generalizability. The examples span diverse scenarios, from portraits to animals and indoor scenes. For all VLMs tested (LlaVA, BakLlaVA, PaLiGemma, and Florence-2), our method successfully localizes objects of interest with high precision. The filtering mechanism proves particularly effective, producing more focused and explainability heatmaps across all models. This is especially evident in the "Table" and "Chair" examples, where the filtered maps more precisely highlight the relevant furniture items while suppressing attention to irrelevant background elements. Notably, while BakLlaVA exhibits relatively sharp localization even without filtering, our approach further refines its attention maps, resulting in more precise object boundaries and reduced noise. The "Horse" example demonstrates how the filtering mechanism helps in distinguishing the subject from the grassy background, producing cleaner attributions across all models. In the "Bicycle" scene, the filtered maps show improved discrimination between the target object and surrounding furniture, highlighting DEX-AR's ability to handle complex indoor environments with multiple objects. The consistency of improvement across different architectures underscores the robustness and generalizability of our filtering approach, while preserving model-specific characteristics in the attention patterns.

**Failure Cases**   Figure 7 presents qualitative examples of failure cases across different VLMs, revealing interesting patterns in their visual reasoning processes. A common pattern observed across all models is the consistent activation of sky and water regions when processing boat-related queries, suggesting the presence of spurious correlations learned during training. For LLaVA, the "train" example demonstrates how the model's attention heavily focuses on the railway tracks rather than the train itself, indicating another potential spurious correlation in the model's learned representations. Similarly, in the "sofa" example, while the model correctly localizes the sofa, it shows high activation across the entire wall area, suggesting an over-reliance on contextual room features rather than specific object recognition. BakLLaVA exhibits an interesting failure mode in the "person" example, where the attribution map reveals that the model's prediction is primarily driven by a face-like drawing on the motorcycle rather than the actual person in the background. This insight helps

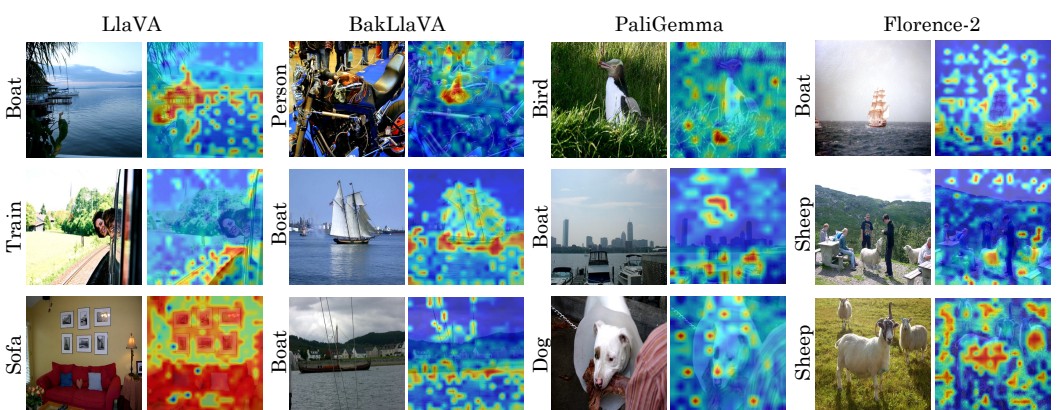

Figure 7: Qualitative examples of failures cases of DEX-AR for different VLMs.

Predicted: The image features a **pile** of **old** and **vintage** items, including a **suitcase**, a **hat**, a **sports ball**, a **clock**, and a **chair**.

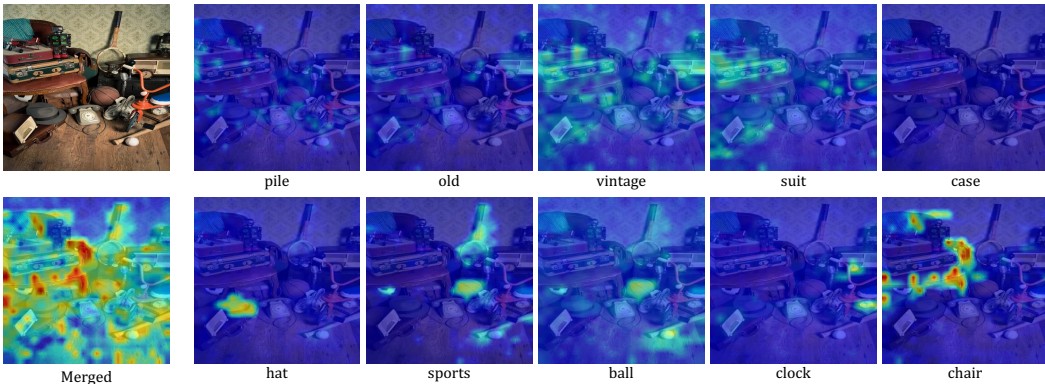

Figure 8: Qualitative evaluation on a complex, cluttered scene. DEX-AR accurately localizes distinct objects (e.g., "hat", "clock") and abstract concepts ("vintage") despite the dense environment. Notably, the method distinguishes between visually grounded tokens (e.g., "suit") and purely linguistic completions (e.g., "case"), effectively suppressing the latter.

explain the model's reasoning process and highlights its tendency to prioritize more prominent face-like patterns, even when they are not the intended subject. PaLiGemma shows sparse and scattered activations for the "bird" example, where the image actually contains a penguin. The attribution map suggests the model's confusion, as it appears to focus on the surrounding grass, possibly attempting to find familiar bird-like features in the environment rather than the subject itself.

Florence-2's behavior on the "sheep" examples is particularly noteworthy, with attribution maps showing strong activation on both the grass and sky regions, despite the sheep being the primary subjects of interest. This pattern suggests the model may be overly reliant on contextual environmental cues commonly associated with sheep in its training data, rather than the distinctive features of the animals themselves. These failure cases collectively highlight how explainability maps can reveal not only where models look but also potential biases and spurious correlations in their learned representations, providing valuable insights for improving model robustness and reliability.

## K QUALITATIVE ANALYSIS ON COMPLEX SCENES

Figure 8 illustrates the performance of DEX-AR on a "hard case" scenario characterized by a cluttered environment containing multiple overlapping objects. The model generates the description: *"The image features a pile of old and vintage items, including a suitcase, a hat, a sports ball, a clock, and a chair."*

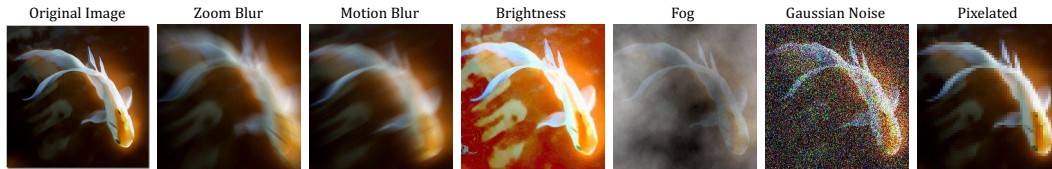

Figure 9: Visual examples of the ImageNet-C corruptions used for robustness evaluation at severity level 5.

**Handling Sub-word Tokenization:** A key insight from this example is the method's ability to distinguish between visually grounded tokens and linguistic completions. The word "suitcase" is tokenized into "suit" and "case". The attribution map for "suit" clearly localizes the pile of suitcases. In contrast, the map for "case" is effectively zeroed out. This confirms that the model predicts "case" primarily based on the preceding linguistic context ("suit"), and our dynamic gating mechanism correctly suppresses this token as it lacks visual sensitivity ($S_{img} < S_{text}$).

**Abstract Concepts and Localization:** The heatmap for the abstract concept "vintage" demonstrates that the model attends to multiple discriminative features simultaneously—highlighting the radios, the leather bag, and the suitcases—rather than focusing on a single object. Furthermore, despite the clutter, the model successfully isolates distinct small objects such as "clock" and "hat" with minimal leakage from surrounding items, demonstrating robustness to occlusion and scene complexity.

## L    FILLER WORDS FILTERING ROBUSTNESS ANALYSIS ON IMAGENET-C

Table 7: Robustness evaluation on ImageNet-C (Severity 5). We report the Signal-to-Noise Ratio (SNR) for different filtering strategies across six corruption types. Higher values indicate better alignment with the target object despite the corruption. Best results are marked in **bold**.

| Filter Setting | Pixelate | Fog | Brightness | Zoom Blur | Gauss. Noise | Motion Blur |
|---|---|---|---|---|---|---|
| **max (Ours)** | **125.90** | **153.53** | 132.02 | **108.97** | **62.72** | **97.72** |
| avg | 7.89 | 9.43 | 9.44 | 5.43 | 1.48 | 6.17 |
| top-$k_{5\%}$ | 118.83 | 140.65 | **135.01** | 103.62 | 47.79 | 97.54 |
| top-$k_{10\%}$ | 103.09 | 128.20 | 122.95 | 90.05 | 30.46 | 76.43 |
| top-$k_{30\%}$ | 96.10 | 149.53 | 137.96 | 62.25 | -15.02 | 48.10 |
| top-$k_{50\%}$ | -1.50 | 7.47 | 10.07 | -10.22 | -16.82 | -9.03 |
| top-$k_{70\%}$ | 1.14 | 2.33 | 2.24 | 0.20 | -3.76 | 1.76 |

To assess the sensitivity of our dynamic head filtering mechanism to outliers and extreme visual conditions, we extended our ablation study to the ImageNet-C dataset (Hendrycks & Dietterich, 2019). We evaluated the method under six distinct corruption types—Zoom Blur, Motion Blur, Brightness, Fog, Gaussian Noise, and Pixelate at the maximum severity level (5). These perturbations introduce significant signal degradation and high-frequency noise, serving as a stress test for the stability of the "max" operation for filtering.

Figure 9 illustrates these perturbations applied to a sample image. We measured the Signal-to-Noise Ratio (SNR) of the resulting attribution maps across different filtering strategies: our proposed maximum ("max"), average ("avg"), and top-$k\%$ normalization at various thresholds.

The quantitative results are reported in Table 7. It shows that the "max" filtering strategy is highly robust, consistently outperforming the 'avg' and top-$k_\%$ strategies. Notably, in the case of **Gaussian Noise**, which explicitly introduces severe pixel-level outliers, the "max" operation maintains a high SNR (62.72) compared to "avg" (1.48) or the best top-$k_\%$ setting (47.79).

This indicates that rather than being sensitive to outliers, the "max" operation effectively acts as a sharp filter: it isolates the specific attention heads that preserve strong connections to the remaining

visual signal, whereas averaging strategies tend to dilute the signal with the noise introduced by the corruption. The "max" strategy achieves the highest SNR in 5 out of the 6 corruption categories, confirming its suitability for processing noisy or degraded inputs in autoregressive VLMs.

sectionRobustness to Vision Transformer Registers

Recent studies have identified a phenomenon in Vision Transformers known as "registers" (Darcet et al., 2024), where the model repurposes certain tokens in low-information background areas (such as corners or plain walls) to store global information. These tokens are characterized by high attention norms, which often mislead attention-based explainability methods into identifying them as salient regions, despite their lack of local semantic relevance to the generated text.

To evaluate DEX-AR's sensitivity to these artifacts, we conducted a qualitative comparison between raw attention maps and DEX-AR heatmaps. As illustrated in Figure 10, raw attention maps frequently exhibit distinct high-norm activations in uninformative regions, which we have highlighted with pink circles. These artifacts typically appear in background areas.

The comparison Figure 10 shows that DEX-AR successfully suppresses these register tokens. While these tokens have high attention magnitudes, their gradient with respect to the specific predicted word is negligible. This indicates that while the model uses these tokens as attention sinks, they do not causally drive the decision for specific semantic tokens (e.g., "cat", "horse"). By relying on the layer-wise gradients of attention maps ($\nabla A^{l,t}$) rather than magnitude alone, DEX-AR naturally filters out these architectural artifacts, resulting in cleaner heatmaps that remain faithful to the visual reasoning process.

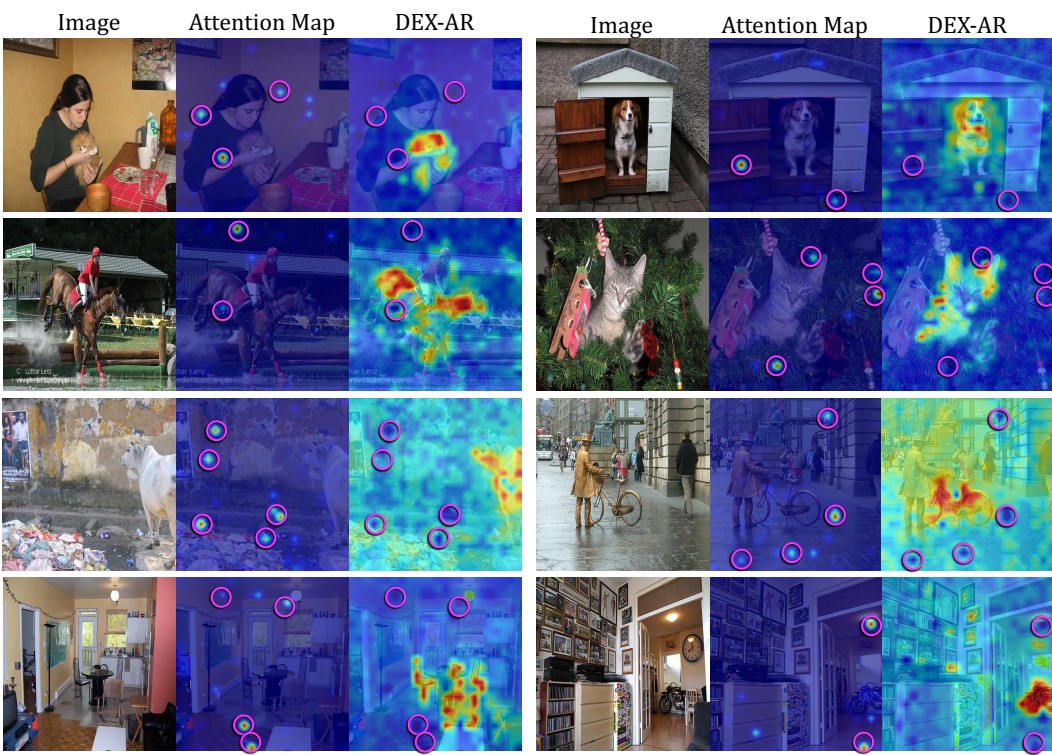

Figure 10: **Robustness to Registers.** Side-by-side comparison of raw attention maps (middle) and DEX-AR heatmaps (right). Raw attention maps frequently suffer from "register" artifacts which correspond to high-norm tokens in uninformative background regions (highlighted by pink circles). DEX-AR effectively suppresses these artifacts, demonstrating that the gradient-based importance score correctly identifies them as irrelevant to the specific token prediction, focusing instead on the semantic subjects (e.g., the cat, the horse rider).

