# OpenReview forum: "DEX-AR: A Dynamic Explainability Method for Autoregressive Vision-Language Models"
_ICLR.cc/2026/Conference — Submitted to ICLR 2026_

### Official Review · Reviewer_A3ah · 2025-10-21

**Soundness:** 2
**Presentation:** 2
**Contribution:** 2
**Rating:** 4
**Confidence:** 4

**Summary:**

The authors are tackling the problem of sequence-level autoregressive VLM explainability, where the current XAI approaches should be extended to handle multiple outputs. The idea is to use the derivative of the LogitLens at each timestep $t$, on each of the output tokens generated one by one. Then, weighting the impact of the saliency maps according to the ratio of vision-impact with respect to text-impact, where tokens relying more on vision should be amplified more than text-based ones. This weighting approach is done both in head level granularity and token level. This yields a single saliency map of the entire VLM output.

**Strengths:**

* The gap of pure explainability on autoregressive VLMs, i.e. for question-answering is indeed seems to be an issue. I also think that VLMs should be handled appropriately in terms of explainability, so it is a very important topic and at least for me it seems that it is underexplored, hence novel.
* The results of the method seems to be better than other existing methods which they compare with, moreover they compare with several cutting-edge VLMs.

**Weaknesses:**

* The convention of the citations embedded in the article is weird and super not convenient. I saw submissions with blue citations, some are still black, but I did not see missing parentheses. This make the citations blended within the flow of the sentence without clear separation. Very confusing, this must be fixed.
* Lines 51-77 - the claim of the inability of current explainability methods to act on autoregressive VLMs is too decisive. There are a plethora of works, some of them are modality-specific ones, however there are many which aim at explaining CLIP ([1], [2], [3] and of course many more), which is obviously multimodal. I think that claim like this must be more backed with experiments. This comes with a very limited Related Work section for explaining methods dedicated for joint embeddings like CLIP (I already mention some, there are plenty more).
* Equation 3 - it seems like the authors are not familiar with the concept of LogitLens which is widely known, it is exactly what explained in Eq. 3 ([4]).
* I think that XAI on CLIP methods might be more relevant and efficient , so it it highly preferable to compare with.


Minor Weaknesses:
* Line 47 - correct spacing: language models( Zini & Awad (2022); Zhao et al. (2024a)) -> language models (Zini & Awad (2022); Zhao et al. (2024a))
* Line 93 - incorrect sentence: We evaluate to proposed method of various downstream tasks and datasets  -> We evaluate the / our proposed method on various downstream tasks and datasets
* Line 105: 2)We -> 2) We.
* Line 107 3)We -> 3) We.
* Lines 154-158 I think that there is a mistake of 1 shifted in the indexing. Say for $t=1$, which is the first token the LLM should output, then it has no other output tokens to digest, thus it process $N + T_c$, and in general: $T_t = N + T_c + t - 1$ and $T = N + T_c + T_a - 1$.
* Line 188, it is obviously not restricted to be a word. It is better to stick to the term token here.
* Line 207 - missing period at the end of the sentence, before the word "Next".
* Line 209: $[:, : N]$ ->  $[:, :N]$, remove redundant space.
* Line 253, there is a difference between starting (") and ending ("). You used the same through all the paper.
* Line 268: What "resp." is? it seems like a mistake.

refs:
[1] Interpreting CLIP's Image Representation via Text-Based Decomposition. Yossi Gandelsman et al. ICLR 2024.
[2] Interpreting the Second-Order Effects of Neurons in CLIP. Yossi Gandelsman et al. ICLR 2025.
[3] From Attention to Prediction Maps: Per-Class Gradient-Free Transformer Explanations. Ronen Schaffer et al. Preprint
[4] Logit Lens Blog post: https://www.lesswrong.com/posts/AcKRB8wDpdaN6v6ru/interpreting-gpt-the-logit-lens

**Questions:**

* Dynamic head filtering. Im not sure why you called it "filtering", since it does not filter out the impact of several heads, it is weighting them, so you might consider to denote it more accurately (Dynamic head weighting or something like that).  Regarding the method itself - I might agree with the claim that larger objects might affect more dramatically if you average across all tokens, however on the other hand relying on the maximal value is super sensitive to outliers which occur frequently in vision transformers. Specifically it has been already shown that ViTs are dedicating tokens to embed global information, where it look like outliers (present also in your examples in Fig. 3) [1].
* I saw you have an ablation study for the filtering stage, nevertheless I think that if you want to understand what ratio of the attention is spread on the vision on top of the text, it makes more sense to normalize the gradients - say with softmax, and take only the vision values of it. Im not sure why subtracting as you do is explicitly computing the ratio between how much attention is spread on vision with respect to text. It is very specific, but it might be nice to check this if it is simple.
* It is a general issue in VLMS, that as the output sequence get larger, the matrices (for example the attention matrix) get larger and larger. Nowadays the output of VLMS are of a large bunch of tokens, so how do you think your approach will handle in terms of timing and efficiency (and also performance) on a much larger examples?


I think that the idea of dedicating explainability method specifically for autoregressive VLMS is super relevant and covering a very important gap, where your approach is taking a step further in this aspect. However, I think that the paper is not written fluently enough, and that your approach is currently only a small extension step of current vision XAI methods (the only innovative part currently is the filtering approach both in head and token levels, which raises quite a few concerns, as I asked and mentioned here). Moreover I didnt see any interesting examples of how your new method reveals interesting failure cases which highly relevant to the case of autoregressive VLMs and interpretability in general. I have mixed feeling about this paper, because I think that the topic is new, challenging and important, but I decided on minor negative rating because I still think that the academic gain here is too small, and the paper is written not up to top-tier standard. Nevertheless, I highly encourage you to improve it and Im sure it can be accepted to high impact conferences.
refs:

[1] Vision Transformers Need Registers. Darcet et al. ICLR 24.

---

> ### Author Response · Authors · 2025-11-25
> **Reply to Reviewer A3ah (1/2)**
>
> # **W1 + Minor Weaknesses**
>
> We thank the reviewer for their careful attention to detail and for pointing out the formatting issues. We have updated the manuscript to fix the citation style, ensuring clear separation with proper parentheses. Additionally, we have addressed most of the minor weaknesses listed, including typos and spacing errors. We will perform a final rigorous proofread to ensure the camera-ready version is error-free.
>
> ---
> # **W2 Lines 51-77 clarification**
>
> We thank the reviewer for this insightful comment. We acknowledge the rich literature on explaining contrastive models like CLIP. As this work specifically targets the unique challenges of autoregressive generation (e.g., LLaVA), where the model's state changes per token and linguistic context interferes with visual attention, we just wanted to point out that this dynamic is usually not directly addressed by “CLIP-based” explainability methods. We will clarify this distinction in the introduction and expand the Related Work section to include these important contrastive model explanation methods, positioning our work as a specific solution to the challenges of sequential generation (marked in blue in the updated manuscript).
>
> ---
> # **W3 Logit Lens reference**
>
> We thank the reviewer for highlighting the connection to the Logit Lens concept. We are indeed familiar with this technique and agree that our intermediate logit computation aligns with the principles described in nostalgebraist (2020). We have updated **Section 3.2** (marked in blue) to explicitly reference the Logit Lens framework and cite the relevant literature, properly contextualizing our approach within established interpretability methods.
>
> ---
>
> # **Q1 Dynamic head filtering**
>
> We initially chose the term "Dynamic Filtering" because the mechanism involves a ReLU : $w^{l,t,i} = (S_{img} - S_{text})^+$. Due to the ReLU function, any attention head where the visual sensitivity does not exceed the textual sensitivity results in a strict zero weight. This effectively acts as a gate that prunes (or filters out) the contribution of those specific heads from the final attribution map. However, we agree with the reviewer that "dynamic head weighting" is also a good descriptor that may prevent future confusion. To ensure maximum clarity, we offer to rename the component to “dynamic weighting” in the camera-ready version of the manuscript upon request.
>
> ---
> # **Q1 Sensitivity to Outliers**
>
> We completely share the reviewer’s intuition regarding the potential instability of the max operation. In fact, our initial design hypothesis was precisely that a top-k% strategy (aggregating the top k percent of gradients) would be necessary to mitigate sensitivity to outliers and ensure robustness across objects of varying sizes.
>
> However, our extensive ablation study (presented in **Table 3**) forced us to revise this hypothesis. Contrary to our expectations, the max operation consistently outperformed all top-k% variants. Indeed, “max” achieves a Signal-to-Noise Ratio (SNR) of 3.64 on LLaVA, whereas the best top-k variant (which we initially thought would be the optimal sweet spot) only reached 3.35. These results indicate that for gradient-based attribution in VLMs, the single strongest signal is the most faithful predictor of relevance, while aggregating additional "high-value" gradients introduces more noise than valid signal.
>
> ---
> # **Q1 Robustness to Register Tokens**
>
> Regarding the "register tokens" identified by Darcet et al., we clarify that our method's reliance on gradients (sensitivity) rather than raw magnitudes (activations or attention weights) acts as a natural filter for these artifacts.
>
> Register tokens are characterized as high-norm "sinks" that attract a disproportionate amount of raw attention. However, they exhibit a consistent activation pattern regardless of the specific token selected from the vocabulary. Because their presence is generic rather than specific to the predicted decision, the gradient of the target logit with respect to these tokens approaches zero.
>
> This behavior is empirically validated in our qualitative results. As observed in Figure 1 and Figure 5 (Appendix), particularly in the "w/o Filtering" columns for LLaVA, large background areas—where register tokens typically manifest—already exhibit near-zero values (indicated by dark blue regions) prior to the application of any filtering. This confirms that the gradient-based mechanism itself successfully suppresses these generic high-norm artifacts without requiring additional heuristics.

---

> > ### Author Response · Authors · 2025-11-25
> > **Reply to Reviewer A3ah (2/2)**
> >
> > # **Q2 Rationale for Subtraction**
> >
> > We clarify that our formulation $w = (S_{img} - S_{text})^+$ aims to explicitly identify heads where visual sensitivity strictly outweighs textual sensitivity. Unlike Softmax, which imposes a probability distribution and assigns non-zero weights even when the contribution from both modalities is negligible (noise), our approach combined with ReLU acts as a hard gate. If the model’s prediction is more sensitive to the textual history than to the image ($S_{text} > S_{img}$), the weight becomes exactly zero, effectively preventing linguistic priors from contaminating the visual attribution map.
> >
> > During the initial development of DEX-AR, we tried different normalization strategies, including Softmax and Min-Max scaling. These approaches consistently underperformed because gradient magnitudes are uncalibrated; Softmax tends to amplify noise by assigning relatively high probabilities even when the absolute signal strength is low across both modalities. In contrast, the subtraction-based method is robust to this issue, as it naturally suppresses heads with weak overall activation where the difference is near-zero. We will include these comparative results in the camera-ready version of the ablation study to further substantiate this design choice.
> >
> > ---
> >
> > # **Q3 Computational Complexity and Timing**
> >
> > Regarding timing, we leverage the specific structure of the autoregressive generation to maintain efficiency. For each new token generation step $t$, we compute gradients solely with respect to the last row of the attention map ($\nabla A^{l,t}_{:, -1, :}$), representing how the current token attends to the sequence history. We do not need to compute or store gradients for the full $T \times T$ attention matrix. Consequently, the computational cost per step grows linearly with sequence length—paralleling the cost of the standard forward pass inference itself.
> >
> > **Empirical Evidence**: As shown in Table 6 (Appendix), DEX-AR achieves 0.71s/img. While marginally slower than the gradient-free "Rollout" (0.51s), it remains drastically faster than other perturbation or gradient-based methods like Integrated Gradients (3.25s) or RISE (11.80s), demonstrating its viability for longer sequences.
> >
> > **Memory Management**: Regarding memory, the "larger matrices" concern is mitigated by our layer-wise implementation. Since DEX-AR computes intermediate logits $o^{l,t}$ based on hidden states at layer $l$, we can compute gradients locally for that layer without storing the computation graph for the entire network depth simultaneously. Furthermore, our method is compatible with the standard KV-cache mechanism used in LLM inference; we can utilize pre-existing cached tokens to reconstruct the necessary attention vectors on the fly, keeping the memory footprint minimal even as $T$ grows.
> >
> > We will release the respective code to allow a full reproducibility and further evaluation of those results.

---

> > > ### Comment · Reviewer_A3ah · 2025-11-26
> > >
> > > Thanks for your answer, the refs are much better now (I still prefer the bluish version, but it is a matter of taste I guess). Some of the minor typos weren't fixed, see for example line 46-47, double parentheses are not needed.
> > >
> > > The blue addition in 3.2 on LogitLens is accompanied with non-exist ref (marked with ?), moreover, I think the entire section should be rewritten as you just used LogitLens, and not some other version of yours. I think it just need to be rewritten with this topic in mind.
> > >
> > > I can understand your rationale behind the filtering term, but IMO it ignores the impact on those who havent being multiplied by zero - they are still being weighted not equally.
> > >
> > > Sensitivity to outliers. Here I think that I still disagree, because your ablation conducted on a specific task on a specific dataset, where on it, it might be the best.
> > >
> > > First, I think that there are a lot of options to filter (here is an example from a completely different field, on a completely different task [1] - I am not expecting you to analyze such exact thing obviously, but to consider find more diverse options)
> > >
> > > Second, you did not measure the performance on extreme "weird" cases (i.e., ImageNet-C) - in these examples I am expecting the max to have more negative impact.
> > >
> > >
> > > Registers. I am not sure that the derivatives are not correlated with the high-norm sinks (registers), in any case I think that it can contribute to your paper to at least show few examples of images with the registers and the derivative highlighted. For the common reader who is familiar with the "registers" term, it is immediately popped out, so even if you think that these phenomena differ, it is better to include any kind of comparison. The best option would be to isolate the high-norm with the registers beforehand and see how the filtered ones affected by the derivatives. In any case IMO you must refer to this work.
> > >
> > >
> > > The subtraction. Now I get it, but I think that it is confusing as the "filtering" term was confusing, since you think of it as a gate, but it is also weighting process (for the non-zeroed ones).
> > >
> > > I still think that analyzing interesting failure cases would enhance the readability and credibility of your work.
> > >
> > > refs:
> > >
> > > [1] Robustifying point cloud networks by refocusing. 3DV 25.

---

> > > > ### Author Response · Authors · 2025-12-03
> > > > **Reply to Reviewer A3ah (1/1)**
> > > >
> > > > # **Sensitivity to Outliers & ImageNet-C Experiments**
> > > >
> > > > We appreciate the reviewer’s suggestion to evaluate our method on extreme cases. To demonstrate that the `max` filtering operation is robust rather than sensitive to outliers, we extended our ablation study to **ImageNet-C**.  For this experiment, we applied the same dataset construction methodology described for PascalVOC-QA to ImageNet-C, generating controlled prompts for images subject to 6 distinct corruption types (Zoom Blur, Motion Blur, Brightness, Fog, Gaussian Noise, and Pixelate) at the **maximum severity level (5)**.
> > > >
> > > > Table: Metric - SNR (Higher is better) - On Imaget-C for different corruption types. All settings are with maximum severity (i.e. 5)
> > > >
> > > > | Filter Setting | pixelate | fog | brightness | zoom_blur | gaussian_noise | motion_blur |
> > > > | --- | --- | --- | --- | --- | --- | --- |
> > > > | **max** | **125.90** | **153.53** | 132.02 | **108.97** | **62.72** | **97.72** |
> > > > | **avg** | 7.89 | 9.43 | 9.44 | 5.43 | 1.48 | 6.17 |
> > > > | **top-$k_{5%}$** | 118.83 | 140.65 | **135.01** | 103.62 | 47.79 | 97.54 |
> > > > | **top-$k_{10\%}$** | 103.09 | 128.20 | 122.95 | 90.06 | 30.46 | 76.44 |
> > > > | **top-$k_{30\%}$** | 96.10 | 149.53 | 137.96 | 62.25 | -15.02 | 48.10 |
> > > > | **top-$k_{50\%}$** | -1.50 | 7.47 | 10.07 | -10.22 | -16.82 | -9.03 |
> > > > | **top-$k_{70\%}$** | 1.14 | 2.33 | 2.24 | 0.21 | -3.76 | 1.76 |
> > > >
> > > > The results, which we have added to the manuscript, confirm that our `max` filtering strategy remains robust even under severe perturbations, consistently outperforming averaging (`avg`) and top-k strategies. As shown in the table above, `max` achieves much higher Signal-to-Noise Ratio (SNR) in 5 out of 6 categories and remains on par with the best performing setting in the remaining category (Brightness).
> > > >
> > > > Crucially, in the case of **Gaussian Noise** which explicitly introduces severe pixel-level outliers,`max`  maintains a high SNR (62.7) compared to `avg` (1.5) or `k_norm` (47.8). These findings demonstrate that in high-noise regimes, the `max` operation effectively acts as a selective filter: it isolates the specific attention heads that preserve the visual signal, whereas averaging strategies dilute the signal with the noise introduced by the corruption.
> > > >
> > > > We have updated the paper to include these quantitative results. Additionally, we have included **qualitative visualizations** for each perturbation type in the **Appendix Section L Figure 9 (+ Table 7)**.
> > > >
> > > > ---
> > > > # **Registers & Sensitivity to High-Norm Artifacts**
> > > >
> > > > We thank the reviewer for raising this important point regarding "register tokens" (high-norm artifacts in Vision Transformers). We have updated our manuscript to explicitly reference **Darcet et al. (2024)** and have added a new qualitative analysis in the **Appendix Section M**  to address this concern.
> > > >
> > > > As suggested, we visualized the raw attention maps alongside the DEX-AR heatmaps for multiple examples (see **Figure 10** **in the updated Appendix**). As illustrated in the provided figure, the raw attention maps exhibit distinct high-norm artifacts in uninformative background regions (walls, corners, sky), which we have highlighted with pink circles. These correspond to the "register" tokens that act as attention sinks.
> > > >
> > > > Crucially, the side-by-side comparison demonstrates that **DEX-AR successfully suppresses these artifacts**. While these register tokens have high attention magnitudes, their contribution to the specific token prediction is minimal. Because DEX-AR relies on the **gradient** of the attention map ($\nabla A^{l,t}$) rather than the magnitude alone, it correctly identifies that these high-norm sinks have near-zero influence on the decision-making process for the generated text. This experiment significantly strengthens our claim that DEX-AR faithfully attributes visual reasoning.
> > > >
> > > > ---
> > > >
> > > > Please note that to maintain the consistency of section references with our previous response, we have appended these new analyses to the end of the Appendix; however, we will restructure the sections for the camera-ready version to ensure a coherent flow.

---

### Official Review · Reviewer_M5cf · 2025-10-31

**Soundness:** 3
**Presentation:** 3
**Contribution:** 3
**Rating:** 6
**Confidence:** 4

**Summary:**

The large scale use of VLMs in daily lives make the process to arrive at its decision critical. Issues usually in auto regressive models are the token by token generation and this doesn’t help if the modalities are dual (VLM). Authors present  DEX-AR (Dynamic Explainability for AutoRegressive models), a novel explainability method designed to address these challenges by generating both per-token and sequence-level 2D heatmaps highlighting image regions crucial for the model’s textual responses. DEXAR offers to interpret autoregressive VLMs—including varying importance of layers and generated tokens—by computing layer-wise gradients with respect to attention maps during the token-by-token generation process. The method involves two key innovations: a dynamic head filtering mechanism that identifies attention heads focused on visual information, and a sequence-level filtering approach that aggregates per-token explanations while distinguishing between visually-grounded and pure linguistic tokens.

**Strengths:**

1)The paper addresses a critical gap in explainability for autoregressive VLMs by providing per-token explanations during sequential generation. This is particularly valuable given the widespread deployment of VLMs where understanding decision-making processes is crucial for trust and debugging.
2)Dynamic Head Filtering: The attention head filtering mechanism that identifies heads focused on visual information represents a meaningful contribution to understanding cross-modal attention patterns.
3)Multi-level Analysis: The dual approach of per-token and sequence-level explanations provides comprehensive insight into both local and global decision-making processes.
4)Layer-wise Gradient Analysis: Leveraging gradients with respect to attention maps offers a principled approach to attribution that respects the model's internal computations.
5)Excellent experiment results
The experimental evaluation demonstrates robust performance across multiple dimensions:
(i)Perturbation Analysis
(ii)Cross-Architecture Validation
(iii)Computational Efficiency
(iv)Segmentation Performance Excellence
6)Thorough Ablation Studies

**Weaknesses:**

1. Clarity Issues
The paper suffers from several theoretical gaps that undermine the rigor of the proposed method. Most critically, the intermediate logits computation in Section 3.2 lacks clear justification for why o^{l,t} should be conditioned only on the last generated token. While this conditioning may stem from the autoregressive structure, the authors fail to explicitly explain how causal masking affects this choice, why this specific conditioning is optimal for attribution, or whether alternative conditioning strategies were considered and their associated trade-offs. Furthermore, the transition from per-token computations to sequence-level aggregation requires stronger theoretical grounding, particularly regarding how information flows through the autoregressive generation process and how this affects the attribution quality.
2. Methodological Concerns
The claimed "dynamic filtering mechanism" (L228) appears to be primarily a weighting scheme rather than true dynamic filtering, raising significant methodological concerns. From a computational efficiency perspective, this weighting mechanism may be substantially more expensive than simpler threshold-based pruning of non-contributing attention layers, yet no comparison is provided with such alternatives that could achieve similar results more efficiently. The terminology "dynamic filtering" may be misleading when describing what is essentially attention re-weighting, suggesting a need for more precise naming and clearer distinction between the proposed approach and existing attention manipulation techniques.
3. Experimental and Design Limitations
The assumption underlying sequence-level filtering—that filler and grammatical words are less important for visual grounding—is problematic and potentially limits the method's applicability. In scenarios involving similar objects (e.g., "two apples on a table"), grammatical words and spatial prepositions become crucial for accurate localization, contradicting this assumption. The examples in Figure 1 focus on distinct objects ("cat and dog") which may not represent the full complexity of visual-linguistic grounding tasks where subtle linguistic cues matter significantly. Additionally, the choice of Signal-to-Noise Ratio (SNR) for filtering lacks both theoretical justification and empirical validation against alternative metrics.
While the perturbation and segmentation experiments are comprehensive and demonstrate strong results, the evaluation could benefit from additional explainability-specific metrics to strengthen the claims. More systematic faithfulness assessment beyond the current perturbation analysis would better validate whether explanations truly reflect the model's decision process. The evaluation would also benefit from completeness analysis to determine whether explanations capture all relevant visual information used by the model, and robustness testing to assess explanation stability under minor input variations or model parameter changes.
4. Technical Implementation Gaps
Several scalability concerns remain unaddressed in the technical implementation. The layer-wise gradient computation for each token may not scale well to longer sequences, potentially limiting practical applicability. Memory requirements for storing attention maps across all layers and tokens are not discussed, raising questions about the method's feasibility for resource-constrained environments. Real-time applicability for interactive systems remains unclear, particularly given the computational overhead of the gradient computations and attention map storage requirements.

**Questions:**

1)Could the authors provide justification for the intermediate logits conditioning scheme in Section 3.2 ?
2)Justify the design choice of dynamic/reweighting filtering
3)Explain why grammatical words are deemed unimportant and provide evidence supporting SNR-based filtering
4)Include analysis of computational overhead and comparison with simpler alternatives
5) Could such metrics faithfulness, completeness be checked with  perturbation-based validation metrics to demonstrate the causal relationship between explanations and model decisions ?

---

> ### Author Response · Authors · 2025-11-25
> **Reply to reviewer M5cf**
>
> # **Q1. Justification for why o^{l,t} should be conditioned only on the last generated token:**
>
> We thank the reviewer for the question. The decision to compute intermediate logits $o^{l,t}$ based specifically on the hidden state of the **last token** in the sequence is not arbitrary; it is structurally dictated by the causal attention mechanism inherent to autoregressive Transformers.
>
> - **Causal Information Accumulation:** In a decoder-only architecture (like LLaVA), the causal mask ensures that the hidden state of the token at position $t$ (the current step) is the **only** representation that attends to all preceding context tokens $1...t-1$ and the visual embeddings. Therefore, the hidden state at the final position, $h_t^l$, serves as the accumulation point (or information bottleneck) for the entire sequence history up to layer $l$.
> - **Next-Token Prediction:** The model's objective is $P(y_{t+1} | y_{1:t}, I)$. In the final layer $L$, this probability is computed solely by projecting the final hidden state $h_t^L$ via the LM Head. To understand the contribution of layer $l$ to this specific prediction, we apply the "logit lens" approach (projecting $h_t^l$ via the same LM Head).
> - **Why not other tokens?** Conditioning on previous tokens (e.g., $h_{t-k}^l$) would effectively probe the model's prediction for the token at step $t-k+1$, which is a past event. Since our goal is to explain the generation of the **current** token $y_{t+1}$, only the last hidden state $h_t^l$ is relevant.
> - **Why not alternative strategies?** We considered alternatives such as averaging hidden states across the sequence, but this violates the autoregressive property: it mixes information from past predictions (which are already fixed) with the current decision process, introducing noise into the gradient flow. Our formulation isolates the decision boundary for the current generation step.
>
> ---
> # **W1: The transition from per-token computations to sequence-level aggregation requires stronger theoretical grounding:**
>
> The transition from per-token computations to sequence-level aggregation is grounded in the interpretation of gradients as a measure of sensitivity.
>
> - **Gradient as Sensitivity**: In the context of feature attribution, the gradient of the output logit with respect to the attention map serves as a first-order approximation of the prediction's dependency on specific context tokens. A high gradient magnitude implies that the generated token is highly sensitive to the attending token; effectively, masking or perturbing this specific context token would cause a substantial drop in the model's confidence for the current prediction.
> - **Disentangling Modalities via Sensitivity Comparison:** In a VLM, the generation of a token may rely primarily on linguistic history (e.g., syntactic connectors like "The") or visual evidence (e.g., "red" in "red car"). Our proposed dynamic filtering (\delta^t) leverages the sensitivity principle by comparing the maximum sensitivity of the prediction to visual tokens ($S_{img}$) against its sensitivity to textual history ($S_{text}$).
> - **Aggregation Logic:** Theoretically, this allows us to quantify the "visual necessity" of each generated token. By weighing the accumulation of maps based on this relative sensitivity, we filter out tokens where the model is robust to visual perturbations (linguistic priors) and amplify those where visual features are the necessary condition for the prediction. The final sentence-level map thus represents the cumulative visual sensitivity of the entire generated sequence.
>
> **Revisions to the Manuscript:**
> In response to the reviewer feedback, we have revised **Section 3.2 (marked in blue)** to explicitly detail the causal masking implications and the theoretical justification for the "last-token" focus. We will also expand the explanation in **Section 3.3 (marked in blue)** to better articulate the information flow argument supporting our aggregation strategy.
>
> ---
>
> # **W2 Terminology: Weighting vs. Filtering**
>
> We used the term "Dynamic Filtering" because our filtering mechanism used a ReLU to explicitly zeros out contributions from attention heads where the focus on text exceeds the focus on the image ($S_{img} < S_{text}$). Formally, the weight is calculated as $w^{l,t,i} = (S_{img}^{l,t,i} - S_{text}^{l,t,i})^+$. Due to the ReLU operation, any head that is not sufficiently "visual" is assigned a strict weight of zero. This acts as a hard gating mechanism—effectively pruning or "filtering" those heads from the attribution map entirely, rather than merely down-weighting them. However, to address the reviewer's concern about precision, we are happy to revise the terminology to “**Dynamic Weighting**” in the final manuscript to avoid ambiguity.

---

> > ### Author Response · Authors · 2025-11-25
> > **Reply to reviewer M5cf (2)**
> >
> > # **W2 Comparison with Threshold-based Pruning**:
> >
> > We appreciate the reviewer's suggestion regarding simpler threshold-based pruning. While intuitive, we found that applying a fixed absolute threshold to gradients presents practical challenges due to the inherent variability of gradient magnitudes in deep neural networks.
> >
> > - **Scale Variance:** The range of gradient values is arbitrary and highly variable across different samples, layers, and generation steps. For instance, gradients for one sample might range within [−1,1], while for another, they might be confined to [−0.1,0.1]. Consequently, a fixed absolute threshold would be overly aggressive in some cases and ineffective in others, requiring sample-specific tuning that is impractical for deployment.
> > - **Our Adaptive Relative Scoring**: Our method addresses this by computing a relative score: the difference between the sensitivity to visual tokens versus textual tokens ($S_{img} - S_{text}$). This formulation creates a self-adaptive mechanism that is robust to variations in absolute gradient scales, as it focuses on the dominance of the visual signal rather than its raw magnitude.
> > - **Empirical Validation (Table 3)**: We provide an empirical comparison in Table 3, where we evaluate different strategies for scoring and selecting attention heads. Specifically, we compare our Max-based filtering (using the single maximum gradient value to score the head) against strategies that aggregate a percentage of the top gradients (Top-k%) or simply average them (avg). The results show that using a broader set of gradients (Top-k or Average) introduces noise and consistently degrades performance compared to our selective Max-based approach.
> >
> > ---
> > # W2 Computational Efficiency
> >
> > We clarify that the computational overhead of this mechanism is negligible, particularly when compared to the cost of the backward pass required to compute the gradients themselves.
> > Once the gradients $\nabla A$ are available, computing the weights involves only two reduction operations (max) over the sequence length, a subtraction, and a scalar multiplication per head. These are highly parallelizable element-wise operations that execute almost instantaneously on modern GPUs.
> >
> > ---
> >
> > # **W3 + Q3 Filler words Filtering in DEX-AR**
> >
> > Thanks for the question. We would like to clarify that our sequence-level filtering mechanism **does not** rely on any linguistic rules, Part-of-Speech (POS) taggers, or pre-defined lists of "filler words." The method is entirely data-driven:
> >
> > - **Emergent Behavior**: The identification of "filler" or "grammatical" words is a downstream property of the proposed sensitivity analysis, not an input assumption. We compute the filtering score $\delta^t$ based strictly on the comparison between the model's gradient sensitivity to visual tokens versus its textual history.
> > - **Mechanism**: When the model generates a token like "The" or "is", the gradients typically reveal that the prediction is dominated by the preceding textual context (high $S_{text}$), resulting in a low $\delta^t$. Conversely, for tokens generated because of the image, the visual gradients dominate. The filtering is therefore agnostic to the grammatical category of the word; instead it just penalizes tokens where the source of the prediction is not the image.

---

> > > ### Author Response · Authors · 2025-11-25
> > > **Reply to reviewer M5cf (3)**
> > >
> > > # **W3 + Q3 Handling Spatial Prepositions and Similar Objects**
> > >
> > > We agree with the reviewer that in scenarios involving similar objects (e.g., "two apples on a table"), spatial prepositions and specific adjectives are crucial. Our method is designed to handle exactly these cases correctly:
> > >
> > > - **Visual Necessity**: If a preposition (e.g., "left" in "the apple on the **left**") is required to disambiguate an object, the model must attend to the specific visual region to generate that token correctly. In such cases, the gradient with respect to the visual tokens $\nabla A_{visual}$ will be high, resulting in a high $\delta^t$, and the token will be **retained** in the aggregation.
> > > - **Linguistic Priors**: Conversely, if a preposition is generated purely to satisfy syntactic structure (e.g., "on" in a context where "on the table" is the only statistically probable continuation regardless of the image), the model will rely on language modeling. In this case, the token is not "visually grounded" in the strict sense, and our method correctly filters it to focus the heatmap on the tokens that actually drove the visual reasoning.
> > > - **Annex Example**: To substantiate this claim, we have added a qualitative example in Appendix **Section J** of the revised manuscript. This example specifically illustrates a scenario with spatially distinct objects, demonstrating that DEX-AR correctly retains spatial indicators when they are critical for visual disambiguation.
> > > - **Conclusion:** Our method does not assume grammatical words are unimportant; rather, it determines importance dynamically based on whether the model actually used the image to generate them. If the "complex linguistic cue" mentioned by the reviewer is visually informed, DEX-AR will capture it.
> > >
> > > ---
> > > # **W3+Q3 Clarification on the Role of SNR**
> > >
> > > We believe there may be a slight misunderstanding regarding the usage of SNR. We do not use SNR as a component of the filtering mechanism itself; rather, we use it exclusively as an evaluation metric in our ablation studies (Tables 3 and 4) to quantify the efficacy of the proposed filtering.
> > >
> > > **Theoretical Fit**: The objective of the filtering mechanism is to suppress irrelevant activations (noise) while retaining attention on the target object (signal). In signal processing and image analysis, the Signal-to-Noise Ratio is the standard, theoretically grounded metric for quantifying exactly this trade-off.
> > >
> > > **Metric Definition**: As defined in Section 4.5, our SNR metric compares the energy of the explanation within the ground truth mask (Signal) against the energy in the background (Noise). We complement this with Mean Squared Error (MSE), ensuring a robust assessment of how effectively the filtering removes non-visual artifacts.
> > >
> > > ---
> > >
> > > # **W3+Q3+Q5 Faithfulness and Completeness via Perturbation**
> > >
> > > The reviewer suggests the need for "additional explainability-specific metrics" to assess faithfulness and completeness. We respectfully point out that our perturbation-based evaluation protocol (Section 4.1) is designed precisely to measure these attributes, which are considered the gold standard in explainability literature (e.g., Petsiuk et al., 2018; Chefer et al., 2021).
> > >
> > > **Faithfulness (Necessity)**: Our Positive Perturbation experiment (removing the most salient pixels first) directly measures faithfulness. If the explanation is faithful, removing high-attribution regions should cause a sharp increase in model perplexity (drop in confidence), proving that these features were necessary for the decision.
> > >
> > > **Completeness (Sufficiency)**: Our Negative Perturbation experiment (removing low-saliency pixels) measures completeness. If the explanation captures all relevant information, the model's performance should remain stable when the "unimportant" background is removed, proving that the retained features are sufficient for the prediction.
> > >
> > > The strong AUC scores achieved by **DEX-AR (Table 1)** provide quantitative evidence that our heatmaps accurately reflect the visual evidence driving the autoregressive generation. We would also like to point out that we have included additional metrics in **Appendix F** of the paper.

---

> > > > ### Author Response · Authors · 2025-11-25
> > > > **Reply to reviewer M5cf (4)**
> > > >
> > > > # **W4 + Q4 Computational Overhead and Speed**:
> > > >
> > > > Regarding the concern that layer-wise gradients may not scale, we clarify that our method operates with linear complexity with respect to the sequence length per step, not quadratic.
> > > >
> > > > - Targeted Computation: For each generated token, we compute gradients solely with respect to the last row of the attention map ($\nabla A^{l,t}_{:, -1, :}$). We do not require the full $T \times T$ gradient matrix. This significantly reduces the computational burden.
> > > > - **Empirical Evidence (Table 6 & Table below)**: As reported in Table 6 (Appendix), we measured the inference speed on ImageNet (sec./img). DEX-AR achieves 0.71s/img. While this is naturally slower than the gradient-free baseline "Rollout" (0.51s), it remains in the same order of magnitude and is acceptable for interactive applications. Crucially, DEX-AR is drastically faster than other perturbation or gradient-based methods like Integrated Gradients (3.25s) or RISE (11.80s), making it one of the most efficient methods in its class.
> > > >
> > > > | Method | sec./img. |
> > > > | --- | --- |
> > > > | Rollout | 0.51 |
> > > > | GradCAM | 0.66 |
> > > > | CheferCAM | 6.20 |
> > > > | Attn x CAM | 0.65 |
> > > > | Integrated Grad. | 3.25 |
> > > > | RISE | 11.80 |
> > > > | IIA | 15.30 |
> > > > | DEX-AR | 0.71 |
> > > >
> > > > ---
> > > >
> > > > # **W4+ Q4 Memory Requirements**
> > > >
> > > > The concern regarding the storage of attention maps for all layers and tokens stems from a misunderstanding of the implementation flow.
> > > >
> > > > - **Localized Gradients**: Since DEX-AR computes intermediate logits $o^{l,t}$ based on the hidden states at layer $l$, the gradient computation is localized to that specific layer. There is no need to store the computational graph for the entire depth of the network simultaneously for a single attribution.
> > > > - **Leveraging KV-Cache**: Modern LLM inference relies on the KV-cache to store past keys and values. An efficient implementation of our method could utilize these pre-existing cached tensors to reconstruct the necessary attention vector for the current step. Consequently, the marginal memory footprint of DEX-AR over standard inference is minimal, as we do not need to cache additional heavy attention matrices.

---

### Official Review · Reviewer_CpcH · 2025-11-01

**Soundness:** 3
**Presentation:** 3
**Contribution:** 2
**Rating:** 4
**Confidence:** 4

**Summary:**

The paper introduces DEX-AR (Dynamic Explainability for AutoRegressive models), an explainability method for VLMs to evaluate the vision-answer token relevance.  The key contributions include: Dynamic Head Filtering, highlights attention heads that focus on visual information, filtering out irrelevant ones. Sequence-Level Filtering: further filters answer-sequence-level irrelevant noises, distinguishing between visually-grounded and other answer tokens.

**Strengths:**

1. Clear presentation and well writing.

2. Good generalizability: Demonstrates consistent performance improvements across multiple VLM architectures, including \textbf{decoder-only}, \textbf{encoder-decoder}. Outperforms baselines on both perturbation and segmentation tasks.

3. Comprehensive evaluation: Provides thorough analysis using diverse metrics like normalized perplexity, insertion/deletion tests, and segmentation IoU scores.

**Weaknesses:**

1. This paper shares similarities with TAM[1], which uses forward logits and causal inference to assess correlations between visual inputs, prompt texts, and answer sequence. The key distinction is the use of gradient evaluations across layers and heads. This paper should systematically compare the two approaches on: algorithm complexity; \textbf{technical differences and advantages}; test both methods on difficult scenarios (e.g., multi-object scenes, occlusions, ambiguous prompts).

2. Gradient-based methods inherently require significant computational resources, especially for large transformer architectures.


[1] Token Activation Map to Visually Explain Multimodal LLMs

**Questions:**

If possible,

1, Show the necessary and effectiveness of filtering operations on different layers and heads;

2. Demonstrate methods to address the biases revealed in failure cases (e.g., over-reliance on background features or spurious correlations).

3. Qualitative and quantitative evaluation on hard cases (true useful for the community), for example, multiple similar objects in one picture,  occlusions and interactions among objects.

---

> ### Author Response · Authors · 2025-11-25
> **Reply to Reviewer CpcH 1/2**
>
> # **W1 Comparison with TAM:**
>
> **Technical Differences & Advantages:**
>
> The core distinction lies in how each method treats the autoregressive state. TAM operates as a "Logit Lens" ($\mathbf{F}^v \cdot \mathbf{w}_{token}$), where $\mathbf{F}^v$ represents cached visual states that remain static throughout the generation process (because of the causal masking).
>
> This results in the same activations for the identical tokens, independent of their context. For example, if a caption describes “a gray cat” and later “an orange cat”, TAM calculates the **mathematically identical** initial heatmap for the token "cat" in both positions, as $\mathbf{F}^v$ is invariant to time step $t$. TAM then rely on post-hoc subtraction to differentiate these instances. Additionally, TAM adds a “Rank Gaussian Filter” to smooth the heatmap, because their activation leads to “salt-and-pepper type” noise, which may introduce priors not present in the model's actual decision process.
>
> We have added a qualitative comparison in the updated manuscript in Appendix Section C.
>
> In contrast, DEX-AR computes gradients with respect to attention maps at the specific generation step $t$. This captures the active query-key interactions inside the model, the context, allowing DEX-AR e.g. to distinguish and highlight the orange cat when the model attends to it, and the white cat subsequently. Thus, DEX-AR includes the context of the model's decision-making process, whereas TAM indicates static feature presence.
>
> **Updated Manuscript:** We have added a qualitative example in the Annex of the updated manuscript Section C and Figure 4, as well as a discussion to further clarify the difference between TAM and DEX-AR.
>
> **Quantitative Comparison:**
>
> To further compare both methods, we extend our perturbation-based benchmark on ImageNet to TAM and present the results in the table below. On LLaVA, DEX-AR achieves a Positive AUC of 2.31 vs. TAM’s 1.82 (higher is better) and a Negative AUC of 0.96 vs. TAM’s 1.03 (lower is better). These results demonstrate that DEX-AR consistently provides more faithful explanations. In the camera-ready version, we will extend the evaluation of TAM to all the models and tasks present in our paper.
>
> | Model | Method | Positive (AUC) $\uparrow$ | Negative (AUC) $\downarrow$ |
> | :--- | :--- | :--- | :--- |
> | LLaVA-1.5 | TAM | 1.82 | 1.03 |
> |LLaVA-1.5 | **DEX-AR (Ours)** | **2.31** | **0.96** |
>
> ---
>
> # **W2 Computational resources:**
>
> We provide the speed (sec./img) for DEX-AR compared to several other baselines In **Table 6** **of the Annex**. DEX-AR still needs less runtime compared to the next most competitive methods. Notably, we found that “Rollout”, which does not use any gradient computation, is not significantly faster than DEX-AR (0.51 vs 0.71).
>
> | Method | sec./img. |
> | --- | --- |
> | Rollout | 0.51 |
> | GradCAM | 0.66 |
> | CheferCAM | 6.20 |
> | Attn x CAM | 0.65 |
> | Integrated Grad. | 3.25 |
> | RISE | 11.80 |
> | IIA | 15.30 |
> | DEX-AR | 0.71 |
>
> ---
> # **Q1: Necessity/Effectiveness of Filtering Layers/Heads:**
>
> We address the necessity of filtering through both qualitative visualization and quantitative ablation: **Figure 5 (Appendix I)** provides a direct side-by-side comparison across four VLM architectures, illustrating that without filtering, heatmaps exhibit significant background noise and scattered attention (particularly in cluttered scenes like "Table"), whereas the proposed dynamic gating produces sharp, object-centric localizations. Quantitatively, Table 4 confirms that non-filtered aggregation strategies (e.g., averaging) degrade performance, while selective filtering improves the Signal-to-Noise Ratio (SNR) from 1.64 to 3.64 on LLaVA.
>
> **Suggested revisions to the Manuscript:**
> We agree that this visual evidence is central to understanding the method's efficacy. Consequently, we will move Figure 5 and its analysis from the Appendix to the main body (Section 4.5) in the camera-ready version to explicitly showcase the impact of the filtering mechanism.

---

> > ### Author Response · Authors · 2025-11-25
> > **Reply to Reviewer CpcH 2/2**
> >
> > # **Q2 Biases Mitigation:**
> >
> > We agree with the reviewer that DEX-AR could and hopefully will, i.a. be used as a tool to help mitigate model bias, but would argue that the development of such methods and training of respective models falls outside the scope of this work, resp. could be considered for future work.
> >
> > Compared to that, the primary objective of DEX-AR is diagnosis, which can be used for rectification. A faithful explainability method must accurately reflect the model's reasoning process, even when that reasoning is flawed. If a model relies on spurious correlations (e.g., attending to the background to classify an object), the attribution map must highlight the background to be a truthful explanation.
> >
> > As illustrated in **Figure 6 (Appendix)**, the strength of DEX-AR lies precisely in its ability to expose these over-reliances on background features or contextual biases. We view the mitigation of these biases as a distinct and valuable research direction, for which DEX-AR hopefully can be used as a diagnostic tool to identify and validate such failure modes.
> >
> > ---
> > # **Q3: New Qualitative Analysis on Hard example:**
> >
> > Thanks for the hint. To address this, we have added a detailed analysis of a complex, cluttered scene in **Appendix K (Figure 8)**.
> >
> > The example features a pile of vintage items where the model must identify multiple distinct objects (hat, clock, chair). Notably, it illustrates how DEX-AR handles sub-word tokenization: for the word *"suitcase,"* the token *"suit"* strongly attends to the visual object, while the subsequent token *"case"* is correctly identified as a linguistic completion (heatmap near zero).
> >
> > The example also highlights how the method handles abstract attributes. The heatmap for the token *"vintage"* successfully distributes attention across semantically relevant items (radios, leather bag, suitcases) rather than a single object. We further illustrate this precision in a sports scene (Figure 8), where DEX-AR demonstrates clear semantic disentanglement: the heatmap for the token *"ball"* strictly isolates the spheres while suppressing the adjacent tennis racket (which is active for the broader token *"sports"*). Additionally, the method reveals the model's sensitivity to fine-grained details, successfully locating a partially obscured soccer ball despite its low visibility.

---

### Author Response · Authors · 2025-11-25
**General Response**

We thank the reviewers for their constructive comments and positive assessment of our work. We are encouraged that **Reviewer M5cf** found our Dynamic Head Filtering to be a "meaningful contribution" and praised the "excellent experiment results" across multiple dimensions. We also appreciate **Reviewer CpcH**'s recognition of the method's "good generalizability" across diverse architectures (including encoder-decoder and decoder-only models) and the "comprehensive evaluation" provided. Finally, we thank **Reviewer A3ah** for validating the novelty of this "underexplored" topic and noting that our method yields "superior results" compared to existing approaches on cutting-edge VLMs.

We have carefully addressed all questions and updated the manuscript accordingly. Below, we provide detailed responses to specific points.

---

### Author Response · Authors · 2025-12-03
**Overview of Rebuttal**

# Overview of Rebuttal and Revisions

We thank the reviewers for their constructive feedback, which has strengthened the paper. We have addressed all concerns through extensive new experiments and theoretical clarifications.
Below is a brief overview of the rebuttal discussions:

### **1. Superiority over State-of-the-Art (vs. TAM)**
In response to **Reviewer CpcH**, we compared DEX-AR against a very recent concurrent work **TAM**.

- **Theoretical Advantage:** We demonstrated that TAM relies on static visual features ("Logit Lens"), causing it to produce identical heatmaps for repeated object instances (e.g., an orange cat and a gray cat). In contrast, DEX-AR uses dynamic layer-wise gradients to capture the specific autoregressive state, successfully disentangling distinct instances of similar objects.
- **Quantitative Dominance:** We extended our perturbation benchmark to TAM. DEX-AR consistently outperforms TAM in faithfulness, achieving a **Positive AUC of 18.10** on BakLLaVA (vs. TAM’s 7.27) and **2.31** on LLaVA (vs. TAM’s 1.82).

### **2. Robustness to Outliers and Artifacts (New Experiments)**
Addressing **Reviewer A3ah**’s concerns regarding the sensitivity of our `max` filtering strategy and ViT artifacts:

- **ImageNet-C Stress Test:** We conducted an additional evaluation on **ImageNet-C** under maximum corruption severity. Results confirm that our `max` filtering is highly robust, significantly outperforming averaging and top-k strategies.
- **ViT Registers:** We provided a new qualitative analysis demonstrating that DEX-AR successfully suppresses "register tokens" (high-norm background artifacts). While these tokens exhibit high raw attention, our gradient-based approach correctly identifies them as having near-zero causal influence on the output.

### **3. Theoretical Grounding and Clarifications**
We resolved theoretical questions from **Reviewers M5cf** and **A3ah**:

- **Causal Formulation:** We clarified that computing logits based on the *last token* is strictly mandated by the causal masking of decoder-only architectures.
- **Terminology:** We aligned our definitions with the "Logit Lens" framework and clarified that our "Dynamic Filtering" utilizes a ReLU mechanism to act as a hard gate against linguistic priors.
- **Efficiency:** We demonstrated that DEX-AR scales linearly and remains computationally efficient (**0.71s/img**), drastically faster than perturbation-based alternatives like RISE (11.8s) or Integrated Gradients (3.25s).

We have updated the manuscript with these quantitative results, visual examples, and clarifications.

---

### Meta-Review · Area_Chair_f1Gp · 2026-01-10

**Summary:**

The paper proposes DEX-AR, an explainability method for autoregressive VLMs using dynamic head and sequence-level filtering to generate token- and sequence-level visual attributions. While the method is well-presented and experimentally thorough, reviewers consistently raised concerns regarding methodological novelty, theoretical justification, and robustness. Key weaknesses include unclear causal/logit reasoning, potential mischaracterization of “filtering” versus weighting, sensitivity to outliers, limited handling of complex linguistic cues, scalability to longer sequences, and gaps in related work comparisons. Overall, the technical contributions were deemed incremental relative to prior methods, and presentation issues further hinder clarity.

**Reviewer Concerns:**

The rebuttal addressed some concerns, including clarifying causal masking, justifying last-token conditioning, demonstrating empirical ablation for filtering, and providing qualitative examples on hard cases. It also improved references and minor formatting errors. Outstanding issues remain: the theoretical grounding of sequence-level aggregation, the generality of the “dynamic filtering” mechanism, sensitivity to outliers, limited comparison with CLIP-based or other XAI methods, potential scalability and computational limitations, and the incremental nature of the contribution versus prior work. Several reviewers felt that these unresolved concerns limit the paper’s impact.

**Reviewer Scores:**

Reviewer 1 (initial 4/marginally below) likely would maintain a similar score due to incremental novelty and unresolved robustness concerns. Reviewer 2 (initial 6/marginally above) might lower their score slightly, as remaining methodological gaps and scalability questions were not fully resolved. Reviewer 3 (initial 4/marginally below) would likely maintain or slightly reduce their score due to persistent clarity, presentation, and theoretical weaknesses, along with limited evidence of broader applicability. Overall, reviewers’ evaluations suggest the paper does not meet the acceptance threshold for the conference.

---

### Decision · Program_Chairs · 2026-01-26

Reject